



# Dating of an East Antarctic ice core (GV7) by high resolution chemical stratigraphies

1 Raffaello Nardin[1], Mirko Severi[1,2*], Alessandra Amore[1], Silvia Becagli[1,2], Francois Burgay[3,4], Laura Caiazzo[5], Virginia Ciardini[6], Giuliano Dreossi[2,3], Massimo Frezzotti[7], Sang-Bum Hong[8], Ishaq Khan[3], Bianca Maria Narcisi[6], Marco Proposito[6], Claudio Scarchilli[6], Enricomaria Selmo[9], Andrea Spolaor[2,3], Barbara Stenni[2,3], Rita Traversi[1,2]

[1]Department of Chemistry "Ugo Schiff", University of Florence, Florence, 50019, Italy
[2]Institute of Polar Sciences of the National Research Council of Italy (ISP-CNR), 30172, Venice, Italy
[3]Department of Environmental Sciences, Informatics and Statistics of the Ca' Foscari University of Venice, Venice, 30172, Italy
[4]Laboratory of Environmental Chemistry (LUC), Paul Scherrer Institut, 5232 Villigen PSI, Switzerland
[5]National Institute of Nuclear Physics (INFN), Florence, 50019, Italy
[6]ENEA, Laboratory of Observations and Measures for the environment and climate, 00123 Rome, Italy
[7]Department of Science, Geologic Sciences section, Roma 3 University, 00154 Rome, Italy
[8]Division of Glacial Environmental Research, Korea Polar Research Institute (KOPRI), Incheon 21990, Korea
[9]Department of Chemistry, Life Sciences and Environmental Sustainability, University of Parma, Parma, 43121, Italy

*Correspondence to: Mirko Severi (mirko.severi@unifi.it)

## Abstract

Ice core dating is the first step for a correct interpretation of climatic and environmental changes. In this work, we release a stratigraphic dating of the uppermost 197 m of the 250 m deep GV7(B) ice core (drilling site, 70°41' S, 158°52'E, 1950 m a.s.l.) with a sub-annual resolution. Chemical stratigraphies of $NO_3^-$, MSA (methanesulfonic acid), non-sea salt $SO_4^{2-}$, sea-salt ions and the oxygen isotopic composition ($\delta^{18}O$) were used in the annual layer counting upon the identification of a seasonal profile in their records. Different procedures were tested and thanks to the volcanic history of the core, obtained in previous works, an accurate age-depth correlation was obtained for the period 1179-2009 CE. Once the dating of the core was finalized, the annual mean accumulation rate was evaluated throughout the analyzed 197 m of the core, obtaining an annually resolved history of the snow accumulation on site in the last millennium. A small, yet consistent, rise in accumulation rate was found for the last 830 years since the middle of the 18[th] century.

## 1 Introduction

Ice cores represent remarkable natural archives able to provide paleoclimatic and paleoenvironmental information, and their study is of high relevance in order to improve our understanding of the climate system. Ice cores are to this day one of, if not the most, valuable archive to obtain long term, highly resolved records of the atmospheric composition and of the





temperatures of the past, spanning from few years up to hundreds of thousands of years (Abram et al., 2013; Delmonte et al.,
2002; Fischer et al., 2007; Traversi et al., 2012; Watanabe et al., 1999; Wolff et al., 2010). Antarctica and the surrounding
ocean play a critical role in climate dynamics (Bertler et al., 2011), but despite the huge efforts of internationals programs
(e.g., ITASE, EAIIST), a large part of the Antarctic ice sheet is still unexplored and additional cores are needed to properly
reconstruct the past climate and to incorporate this information in climate modeling simulations. In particular, the last
millennium is a critical time frame for putting the more recent human related climate change into a longer temporal context
and to disentangle natural versus human impacts on climate variability, but it is still poorly investigated, particularly in
Antarctica. New ice core records from Antarctica are needed for a better assessment of the surface mass balance (SMB) of
the Antarctic continent, which is highly relevant to understand its role in sea-level rise in recent decades and in the near
future (DeConto and Pollard, 2016; Krinner et al., 2007). Spatial coverage of climatic observation in Antarctica and the
Southern Ocean is still poor (Jones et al., 2016; Neukom et al., 2018) and ice core records have the potential to investigate
past SMB beyond the instrumental and satellite period. Recently, Thomas et al., (2017) investigated the Antarctic snow
accumulation variability over the last millennium at regional scale using a large number of ice core snow accumulation
records, grouped and assigned to different regional Antarctic areas and compared with modeled SMB.
In the framework of the PNRA project "IPICS – 2kyr-IT", representing the Italian contribution to the project "The IPICS 2k
Array: a network of ice core climate and climate forcing record for the last two millennia", the latter being one of the four
topics of the International Partnerships in Ice Core Sciences (IPICS), several drillings have been carried out in the Oates
Coast, East Antarctica. In this framework the site named GV7 (Figure 1) was chosen to retrieve ice cores covering at least
1000 yr of climatic and environmental history of this area of Antarctica. The drillings were accomplished through a bilateral
Italy – South Korea collaboration, during the 2013/2014 Antarctic summer.
One of the most critical aspect of the study of the ice core records is the dating of each ice layer, which is fundamental to put
the records into a temporal scale. Different methods were developed since the second half of the last century (Hammer,
1980) including the identification of seasonal pattern in chemical and physical stratigraphies (Alley et al., 1997; Cole-Dai et
al., 1997; Extier et al., 2018; Sigl et al., 2016), ice flow models and identification of temporal horizons such as volcanic
eruptions that brings spikes in the acidity of an ice layer and/or trace elements concentration (Castellano et al., 2005; Igarashi
et al., 2011).
Here, we focused on the identification of seasonal patterns in the ionic and isotopic composition of the core, the latter being
one of the most reliable and extensively used method used to date many ice cores. Since $\delta^{18}O$ in falling snow varies with
seasons (Dansgaard, 1964), showing maxima in summer and minima in winter, it is possible to identify an annual cycle in
$\delta^{18}O$ which is useful in the dating of a core. A similar annual pattern with either summer or winter maxima is found in both
major sea salt and non-sea salt ions found in ice cores. Both methanesulphonic acid (MSA), nitrate and the non-marine
fraction of the sulphate ($nssSO_4^{2-}$) have a seasonal pattern that could be used in the ice core dating (Pasteris et al., 2014;
Piccardi et al., 1994; Stenni et al., 2002; Udisti, 1996). MSA and $nssSO_4^{2-}$ mainly arise from the atmospheric oxidation of
their precursor dimethyl sulfide (DMS), which in turn is produced by metabolic activities of marine phytoplanktonic species



(Stefels et al., 2007). The strong seasonality of DMS production leads to an analogous seasonal behavior of $nssSO_4^{2-}$ and
MSA with the highest concentration peaks during the phytoplanktonic bloom, occurring in austral spring-summer
(November-March) (Becagli et al., 2012).
Unlike MSA, which only arises from marine DMS (Gondwe et al., 2003), $nssSO_4^{2-}$ is formed also from the oxidation in
troposphere of $SO_2$ (Delmas et al., 1985), emitted during explosive volcanic eruptions, to sulphuric acid. Such acid
components thanks to tropospheric and stratospheric circulation (Delmas et al., 1985) could deposit in the Polar regions
during a period of 2-3 years after the event (Sigl et al., 2015) and their signal is superimposed over the biogenic background
of the $nssSO_4^{2-}$. The identification of such volcanic signatures in ice core records is commonly used to synchronize ice core
timescales (Severi et al., 2007, 2012; Winski et al., 2019) and widely used to assign an absolute date to ice layers in a core
(Castellano et al., 2005; Sigl et al., 2013) in conjunction with the annual layer counting.
The same seasonality (with a maximum in the austral summer) is also noticeable in the nitrate concentration throughout the
year. As one of the most abundant ions found in the cores (Legrand et al., 1999), nitrate is considered the final sink of
atmospheric $NO_x$ and thanks to its role and how it interacts with the main oxidant cycles in the atmosphere is considered one
of the potential markers to reconstruct the oxidizing capacity of the past atmosphere (Dibb et al., 1998; Hastings et al.,
2005). These oxidizing processes, combined with the photochemical ones, occur more intensely during summer (Erbland et
al., 2013; Grannas et al., 2007) and lead to a seasonal behavior of this marker as found in polar records (Stenni et al., 2002;
Wolff, 1995). Such clear annual cycles have been used as components of the layer-counted dating of ice cores (Wolff, 2013)
from both hemispheres (Rasmussen et al., 2006; Thomas et al., 2007). Since major sea-salt ions show late winter maxima in
their concentrations in the innermost regions of Antarctica (Udisti et al., 2012) due to large influx of sea salt aerosol during
winter months (Bodhaine et al., 1986), their chemical stratigraphies too could be used for the dating of the core. The same
summer minimum and winter-spring maximum pattern was observed at coastal stations (Mulvaney and Wolff, 1994; Weller
et al., 2011) and $Na^+$ and $Mg^{2+}$ stratigraphies were successfully used in the dating of ice cores (Herron and Langway, 1979;
Winski et al., 2019). Here we present the dating of the uppermost 197 m of the 250 m deep ice core collected at GV7,
focusing on the interpretation of ionic stratigraphies of the core and investigating which one could be best suitable for the
dating itself with support of the high resolution δ18O data available for the first 38 m of the core. Volcanic tie points and
annual layer counting were both used in order to assign an absolute date to the layer and a relative, in-between layer, dating
to the ice. Different dating procedures were tested and the most reliable were used concurrently with the volcanic record.
Once the dating for the ice core was finalized, the snow accumulation rate at the site was evaluated.

## 2 Materials and Methods

### 2.1 Sampling site

The GV7 drilling site is in the Oates Coast, a coastal area of the East Antarctica. The site was chosen for its relatively high
snow accumulation rate (241± 13 mm w.e. yr$^{-1}$ over the past 50 years), the thickness of the ice (approx. 1700m), the limited





post depositional processes due to the reduced force of katabatic winds along the ice divide (Becagli et al., 2004; Frezzotti et al., 2007; Magand et al., 2004) and the excellent stratigraphy (chemical and isotopic) (Caiazzo et al., 2017; Delmonte et al., 2015; Frezzotti et al., 2007). Internal layers of strong radar reflectivity observed with ground-penetrating radar (GPR) are isochronous, and surveys along continuous profiles provide detailed information on the spatial variability of snow accumulation. Spatial distribution of snow accumulation from GPR layer (dated to 1905 ± 9 AD) has been conducted during the 2001-2002 ITASE expedition from 150 km north of GV7 up to Talos Dome (Frezzotti et al., 2007). Spatial distribution of snow accumulation from GPR layer upstream GV7 site shows that internal layering and surface elevation are continuous and horizontal up to 10 km from the site, revealing low ice velocity 0.3±0.01 m yr$^{-1}$, no distortion of isochrones due to ice flow dynamics and very low snow accumulation spatial variability (less than 5%, Frezzotti et al., 2007). An extensive chemical dataset covering 7 years of deposition on site obtained from the analysis of two snow pits is already available (Caiazzo et al., 2017), as well as an in-depth reconstruction of the past volcanic history (Nardin et al., 2020).

**2.2 Fieldwork**

During the 2013/14 Antarctic campaign, six shallow firn cores (5 to 50 m) and two intermediate firn-ice core (87 and 250 m deep) were retrieved. The 250 m deep core (named GV7(B) is the object of this work. The drilling of the ice core (see Figure 1) was accomplished using an electromechanical drilling system (Eclipse Ice drill Instrument).

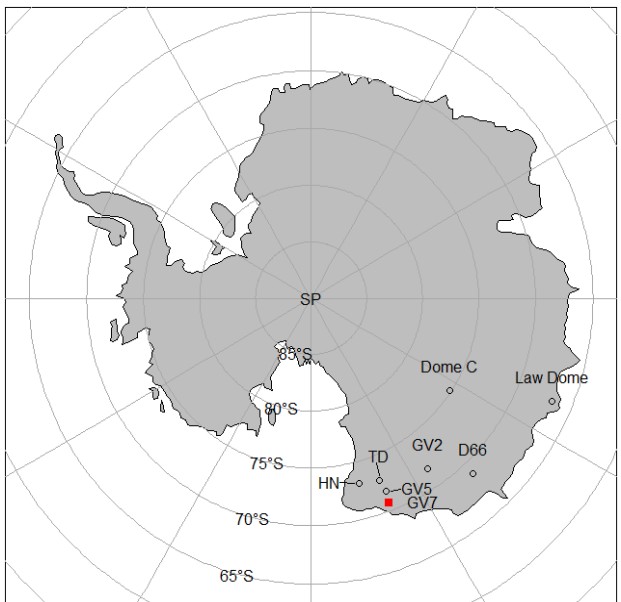

**Figure 1: GV7 (70°41'17.1'' S, 158°51'48.9'' E; 1950 m) drilling site (red square). Hercules Nevè (HN), Talos Dome (TD), GV5, GV2, D66, Dome C and Law Dome ice cores' drilling site are also reported.**

The drilling started at 3 m from the snow surface and reached a depth of 250.2 m. Drilling fluid (Exxsol D40) was used from a depth of 80 m (close off 75 m) and inserted inside the hole with a tube. A level of 4 m of fluid was found to be ideal to aid





the drilling operation and improved the quality ice cores. The Eclipse system has experienced problems during the drilling
below 100 m of depth, the brittleness of the ice, breaks in the core and the presence of drilling fluids in these cracks proved
to be a problem in the decontamination of the core. The presence of numerous breaks and therefore a lack in continuity of
the stratigraphy, prevented us to analyze the deeper part of the core and only the first 194 meters (reaching a depth of 197 m)
were analyzed.

**2.3 Ice core analysis**

60 cm long ice core sections (cut and logged directly in field) were shipped to the EUROCOLD lab of the University of
Milano-Bicocca (Italy) where they were cut longitudinally and transversally and distributed among different research groups.
The 4x4 cm core strips for ionic content analysis were sealed in plastic bags, shipped frozen to the cold room of the
Department of Chemistry of the University of Florence (Italy) and stored at -20°C until the moment of analysis. Conversely,
both the bag (60 cm) and the high-resolution samples (4 cm) for the isotopic analysis were melted at room temperature and
transferred in 25 mL HDPE bottles at the EUROCOLD lab and then sent to the Ca' Foscari University of Venice and the
University of Parma for the isotopic measurements.
The strips for ionic content were manually decontaminated inside the cold room of the Department of Chemistry of the
University of Florence (Italy), by scraping the outermost layer of ice (approx. 1 cm) using ceramic knifes to remove external
contamination (Candelone et al., 1994; Chisholm et al., 1995; Tao et al., 2001, Caiazzo et al., 2016).
All decontamination procedures were carried on under a class-100 laminar flow hood and the sub samples were stored inside
pre-cleaned plastic vials and analyzed within a week to avoid external contamination. For those sections of the ice core too
badly damaged to be manually decontaminated (due to problems in the drilling operations, handling and fracturing of the
ice), the fractures were logged, and the sample decontaminated just before the analysis by quickly submerging it three times
(10 seconds the first wash then 5 seconds the remaining two) in ultra-pure Water (18.2 MΩ 25°C) in order to remove the
outer layer of ice. Each sub-sample was melted at room temperature under laminar flow hood just before the analysis. The
sub-samples were then analyzed for ionic content using two Ion Chromatograms operating simultaneously: a Thermo Dionex
ICS  1000 for the determination of the cationic content ($Li^+$, $Na^+$, $NH_4^+$, $K^+$, $Mg^{2+}$, $Ca^{2+}$) and Thermo Dionex DX  500
equipped with a GP50 gradient pump for anionic content ($F^-$, Formate, methanesulfonate ($MS^-$, referred in the text as MSA),
$Cl^-$, $NO_3^-$, $SO_4^{2-}$). Further details about the columns used and the daily calibration procedures for each ion chromatographic
system are described in (Caiazzo et al., 2016; Morganti et al., 2007), while the analytical performance of the methods were
tested and described in (Nardin et al., 2020).
Samples for isotopic analysis did not require any decontamination procedure. Bag samples (60 cm) were analyzed for $\delta^{18}O$ at
the University of Parma, using a Thermo-Fisher Delta Plus Isotope-ratio Mass Spectrometer (IRMS) coupled with a HDO
automatic equilibration device, following the classical water-$CO_2$ equilibration technique described by Epstein and Mayeda,
(1953). High resolution samples (4cm) were analyzed for $\delta^{18}O$ at the Ca' Foscari University of Venice, using both the IRMS
water-$CO_2$ equilibration technique (Thermo-Fisher Delta Plus Advantage coupled with a HDO automatic equilibration



device) and the Cavity Ring-down Spectroscopy (CRDS) technique (Picarro L1102-I). The Thermo-Fisher Delta Plus and
the Delta Plus advantage are both characterized by an analytical precision of 0.05‰ for $\delta^{18}O$, while the Picarro L1102-I has
an analytical precision of 0.10‰ for $\delta^{18}O$. All measurements were calibrated using internal isotopic standards periodically
calibrated against the certified International Atomic Energy Agency (IAEA) standards VSMOW2 and SLAP2- All the
isotopic data are reported in the SMOW-SLAP δ-scale.

**2.4 Major ions contribution**

Chemical stratigraphies of the relevant ions for the dating procedure were obtained by plotting the concentration (in µg/L)
against the mid depth of the sample, logged during the decontamination procedure. Minimal manipulation was done over the
raw dataset of the different ions' concentration, but extremely high concentration points (i.e. spikes in the concentration of a
single ion) were discarded and attributed to external contamination. This was done by removing all the points above the 99th
percentile for all the ions used in the dating procedure; this was chosen as a good compromise to keep high values of
concentration due to particular events (e.g., volcanic eruptions) and at the same time remove those due to possible
contamination. The non-sea salt sulphate (from now on referred as $nssSO_4^{2-}$) was calculated by equation (1)
$$nssSO_4^{2-} = totSO_4^{2-} - 0.25 * Na^+ \qquad (1)$$
where 0.25 is the average $SO_4^{2-}/Na^+$ ratio in sea water, $totSO_4^{2-}$ and $Na^+$ are the total measured concentration of the two
ions respectively. We assumed that the only contribution for sodium is the sea spray aerosol (Legrand and Delmas, 1984;
Maupetit and Delmas, 1992). Both in inland (Röthlisberger et al., 2002) and coastal sites (Benassai et al., 2005; Nyamgerel
et al., 2020) the crustal contribution of sodium is found to be very low or negligible compared to the marine one. When
calculated for GV7 using a simple equation system (2) (Becagli et al., 2012; Udisti et al., 2012)
$$tot\text{-}Na^+ = ss\text{-}Na^+ + nss\text{-}Na^+ \qquad (2)$$
$$tot\text{-}Ca^{2+} = ss\text{-}Ca^{2+} + nss\text{-}Ca^{2+}$$
$$ss\text{-}Na^+ = tot\text{-}Na^+ - 0.562 \; nss\text{-}Ca^{2+}$$
$$nssCa^{2+} = tot\text{-}Ca^{2+} - 0.038 \; ss\text{-}Na^+$$
where 0.562 and 0.038 represent the $Na^+/Ca^{2+}$ w/w ratio in the crust (Bowen, 1979) and seawater (Nozaki, 1997),
respectively; the non-sea salt contribution of $Na^+$ was found to be 3% as average, lower than the analytical error for ions
determination.
The $nssSO_4^{2-}$ was used in the identification of volcanic signatures in the GV7 ice core using already established methods
(Castellano et al., 2004, 2005; Sigl et al., 2013; Traufetter et al., 2004) on Arctic and Antarctic ice cores, here briefly
described. The biogenic background was calculated as the running average of the $nssSO_4^{2-}$ concentrations and its standard
deviation (σ) was used to set the threshold over which a sample point was to be attributed to a volcanic eruption. Both 2σ



and 3σ were used as thresholds added to the average biogenic background as described more in details in Nardin et al.,
(2020) where an in-depth discussion of the volcanic fluxes of the volcanic eruptions found in the core is also present.

**2.5 Trace element analysis**

The ice samples were analyzed with an Inductively Coupled Plasma Single Quadrupole Mass Spectrometer (ICP-qMS,
Agilent 7500 series, USA) equipped with a quartz Scott spray chamber. A 120-seconds rinsing step with 2% $HNO_3$
(Suprapure, Romil, UK) was performed after each sample to limit any possible memory effect, the vials used for standard
preparation were cleaned following the same procedure adopted for ice samples. The $^{209}Bi$, $^{205}Tl$ and $^{238}U$ quantification was
performed using external calibration curves with acidified standards (2% $HNO_3$, Suprapure, Romil, UK) from dilution of
certified IMS-102 multielemental standard (10 ppm ± 1%, Ultra scientific). The resulting external calibration curves for the
three elements were 0.999. The Limit of Detection (LoD) for Tl and U was 0.001 ppb while for Bi 0.004 ppb, calculated as
three times the standard deviation of the blank.

**2.6 Trend analysis**

Trend analysis of the cores was based on the calculation of breakpoints between periods with significantly different trends
following Tomé and Miranda (2004). The methodology uses a least-squares approach to compute the best continuous set of
straight lines that fit a given time series, subject to a number of constraints on the minimum distance between breakpoints
and, optionally, on the minimum trend change at each breakpoint. We chose a period of 150 yr as minimum distance to
identify trend at seculars scale. The choice is subjective, but it takes into account the high computational request for too
small minimum distance and the risk of non-significance for too large minimum distance. Due possible noise connected to
local spatial variability (Frezzotti et al., 2007) at the three sites (Talos Dome, GV7 and Law Dome) we applied the procedure
to seven years smoothed average in order to make all the cores comparable between each other.

**3 Results and Discussion**

The relatively high snow accumulation rate on the site (well above 200 mm w.e. yr$^{-1}$ Frezzotti et al., 2007) allows an
accurate dating of the core by counting successive snow layers, identifiable by markers having seasonal pattern and/or the
identification of specific dated event, mainly in the form of volcanic eruptions identified in the stratigraphies as spikes of
$nssSO_4^{2-}$ statistically higher than the biogenic background (Nardin et al., 2020).

**3.1 Ice core dating procedure – upper section**

The previous work on snow pit dating at the GV7 site (Caiazzo et al., 2017) revealed that $nssSO_4^{2-}$ and $\delta^{18}O$ stratigraphies
show the best seasonal pattern with summer maxima in phase between them. Therefore, for the uppermost section of the core
(38.27 m) for which the $\delta^{18}O$ high resolution stratigraphy was available, two independent dating using $nssSO_4^{2-}$ and $\delta^{18}O$





respectively were produced assigning to each local maximum of the nssSO$_4^{2-}$ profile the date 1$^{st}$ of January of any given
year. The two records vs. depth were compared and are shown in Figure 2. In both profiles, a clear seasonal pattern can be
identified, and it was used to accurately date the first 40 m of the core. Minor discrepancies between the two profiles are to
be expected and are probably due to the slightly different depth resolution of the two series (4.5 cm and 4 cm on average for
nssSO$_4^{2-}$ and δ$^{18}$O, respectively).

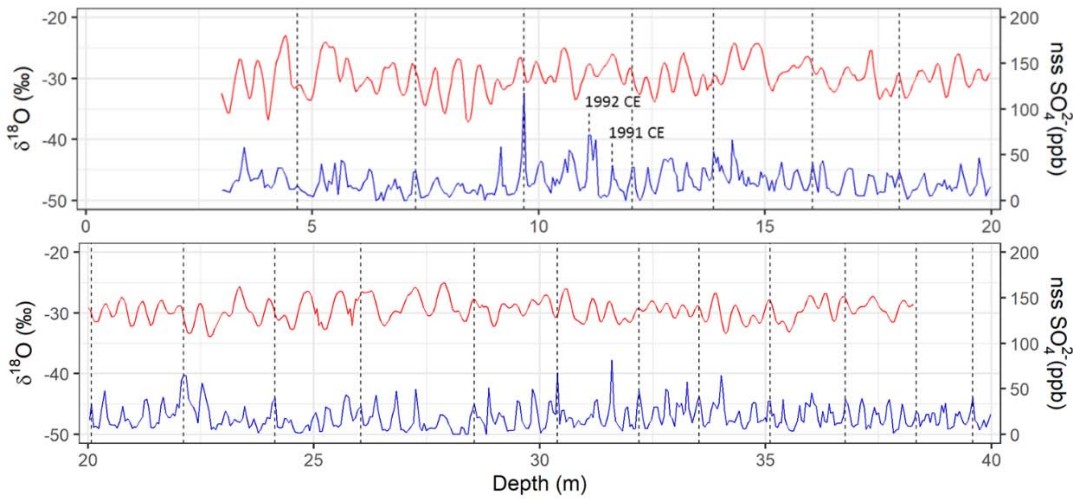


**Figure 2: δ$^{18}$O (red, left scale) and nssSO$_4^{2-}$ (blue, right scale) profiles against depth used in the dating of the uppermost 40 m of**
**the GV7 core. Vertical dashed lines represent intervals of 5 years starting at Year 2005 CE at a depth of 4.67m**
In order to constrain the dating and assign an absolute date to the layer, two volcanic signatures were found in the time
period investigated: 1992 Pinatubo eruption (found at a depth of approx. 11.1 m) and 1964 Agung volcanic eruption (found
at a depth of approx. 22.1 m). As discussed in previous work, neither of these volcanic eruptions display a strong signal in
the nssSO42- in the GV7 core and trace element stratigraphies were used to constrain the 1992 CE tie point identification.
Trace elements deposition in polar ice cap is mainly associated with dust deposition. Evidence for anthropogenic
contribution in the global trace elements deposition is well documented, such as the increase in lead depositional flux in
connection with the introduction of lead containing gasoline. Specific trace elements such as Tl and Bi have been proposed
to be enriched in deposition derived from volcanic eruption (Candelone et al., 1995; Kellerhals et al., 2010). Bi, Tl and U
show an increased concentration between 11.0 and 12.5 m depth (Figure 3), corresponding with the 1989-1992 CE time
period, according to the δ$^{18}$O annual layer counting dating. Bi, Tl and U concentrations peaks, to be attributed to the 1991
CE Cerro Hudson and/or the 1991 CE Pinatubo eruption, are recorded at a higher depth compared to the nssSO$_4^{2-}$ and the
deposition of sulphuric compounds for these eruptions in the Antarctic plateau occurs mainly in the year 1992 CE. However,
dust (and therefore trace elements) deposition occurs earlier, as reported by Hwang et al. (2019) analyzing the same volcanic
eruption from snow pits drilled near Dome Fuji in Dronning Maud Land, hence further consolidating the date attribution to
the trace elements and nssSO$_4^{2-}$ concentration's spikes in this work (1992 CE). Therefore, the uppermost section of the core





(3.00 to 38.27 m) was dated and was found to cover the time period 2009-1920 CE. The uncertainty of this dating is
discussed further below.

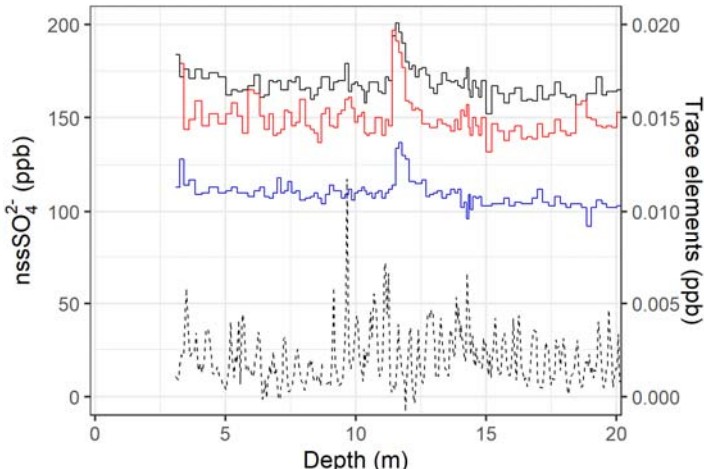


**Figure 3: comparison of the nssSO$_4^{2-}$ (dashed line, left scale) and Tl, Bi, U (black, red and blue line respectively, right scale)
stratigraphies for the first 20 m of the GV7 ice core**

**3.2 ice core dating procedure – lower section**
Due to the lack of high-resolution data for $\delta^{18}O$ for the deeper part of the core, only ion signatures could be used for the
dating of the rest of the core. In order to highlight the seasonal character of each ion and to decide whether or not some ions
are more useful than others for the GV7 ice core dating, we considered the concentration profile of each ion throughout the
89 years already dated in the above section. Each year was equally divided in 4 parts corresponding to the Antarctic seasons
and roughly to the time periods January-March, April-June, July-September and October-December. It has been noticed that
the equal division in four time periods is an approximation as snow deposition is not constant throughout the year in the
coastal regions of Antarctica but considering a dataset of more than 700 sample points spanning a time period of 89 years
this approximation can be acceptable. By dividing each year in just four intervals, we were able to understand which ion(s)
showed the more pronounced seasonal pattern in the core by using bin plots (Figure 4). Both winter and summer maxima are
usable for the annual layer counting procedure. When comparing the profile of each ion to the average calculated in the
considered time interval, both sea-salt ions (Cl$^-$, Mg$^+$ and Na$^+$) and non-sea salt ones (NO$_3^-$, nssSO$_4^{2-}$) showed a maximum
throughout the year, but as shown in Figure 6, the more pronounced one was the one of the nssSO$_4^{2-}$ with most of the lower
concentration points in the 5-10 µg/L bin (middle of the year) and most of the higher concentration points in the 35-40 µg/L
bin (beginning of the year). Sea-salt ions showed winter maxima, and especially the Mg$^{2+}$ profile (Figure 6 c), with generally
higher values of concentration (up to 3.5 µg/L compared to an average of 1.71 µg/L), but in general the most populated bins
in the winter and summer periods showed similar concentrations, suggesting a lack of a pronounced seasonal pattern that
could be helpful in the dating procedure.





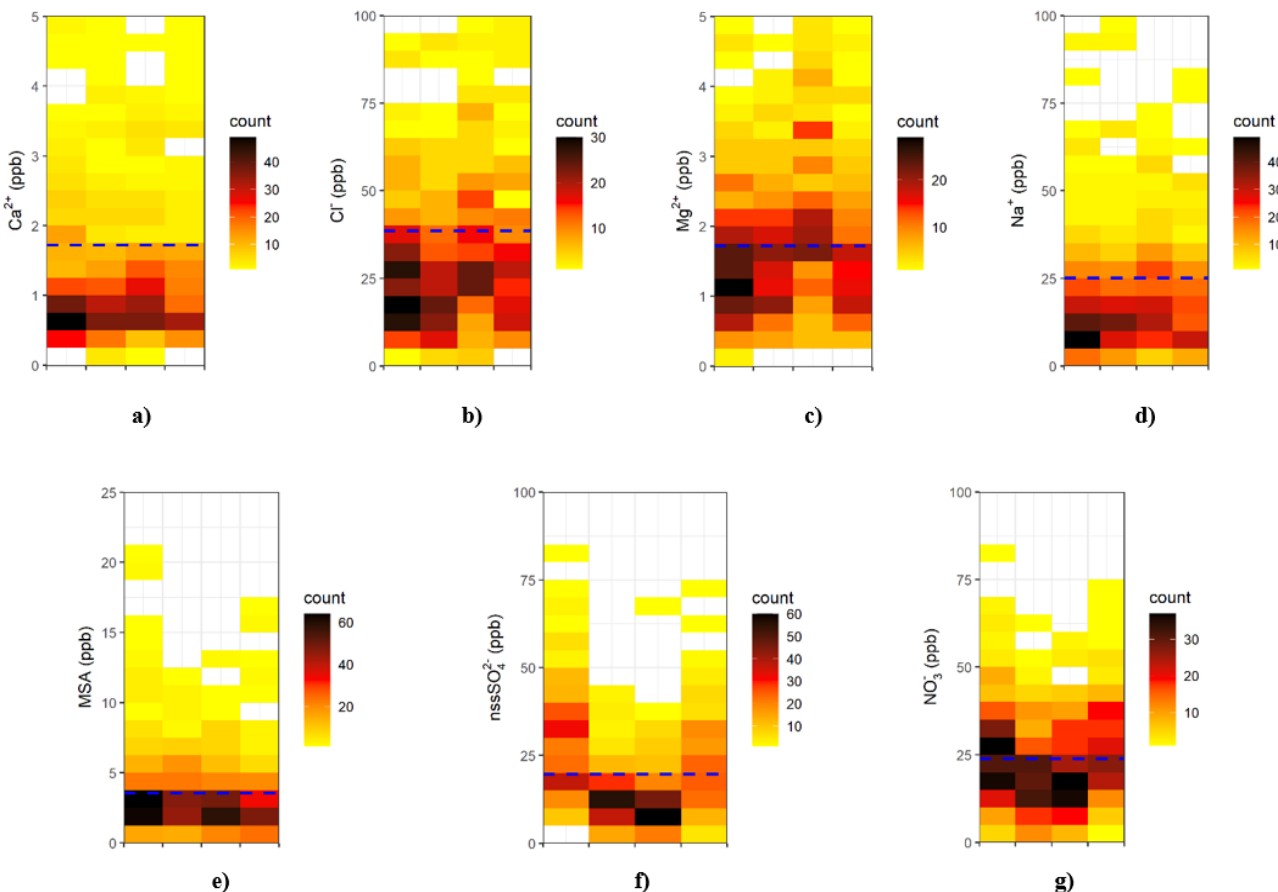

**Figure 4: Seasonal variability of sea-salt ions (Ca$^{2+}$ (a), Cl$^-$ (b), Mg$^{2+}$ (c), Na$^+$ (d)) and non-sea-salt ions (MSA (e), nssSO$_4^{2-}$ (f), NO$_3^-$ (g)) found in the GV7(B) ice core. Concentration's bin are 3 "months" in width and 5 ppb in height except for MSA levels (1 ppb) and Calcium and Magnesium (0.25 ppb). Upper concentration limits and bin sizes were chosen to keep between each ion's plot the same proportions in order to facilitate the interpretation of the data. In blue, the average concentration of each ion in the time interval investigated.**

These considerations only cover a small section of the core (approx. the 20% of its length and due to the compression of the snow and the ice layers, reasonably even less so when considering the time period investigated with this core), but as shown in Figure 5, nor the Na$^+$ nor the Mg$^{2+}$ concentration vs. depth profile seem to show a clear and usable annual pattern at greater depths of the core and not always a maximum in the nssSO$_4^{2-}$ concentration coincide with a minimum in the other two ions concentration. This lack of anticorrelation was further highlighted when a principal component analysis (PCA) was performed on the whole dataset of concentrations measured in the core: by doing so the n-dimensions dataset was reduced to a lower number of dimensions (Principal Components or Factors) by means of orthogonal linear correlation of the interrelated variables The first PC explains the most variation of the n-dimension dataset. PCA was performed using the software STATISTICA (extraction: principal components, rotation: Varimax normalized).

Two factors were extracted with PCA (see Table S1 and Figure S2). As expected, a strong correlation between sea-salt ions was found (grouped in Factor 1), as well as a good correlation between NO$_3^-$ and nssSO$_4^{2-}$ levels throughout the core





(grouped in Factor 2). In Factor 2 $NO_3^-$ and $nssSO_4^{2-}$ present the highest factor loading, but they are negative, highlighting
the opposite seasonal pattern of the two factors: concentration maxima in winter and summer for Factor 1 and 2,
respectively.
Based on the seasonality of the ionic marker highlighted by the bin plots and PCA analysis, a number of dating procedures
reported in literature were checked for GV7 ice core.

1) Multiparametric approach using the sum of MSA, $NO_3^-$ and $nssSO_4^{2-}$ normalized concentrations as reported by
    Udisti (1996). Normalization means that every concentration is divided by the values of the nearest concentration
    maxima, in this way is possible to give the same relevance to maxima having different concentration values in the
    same data series and from different data series.

2) Since by normalizing the three series (all showing similar patterns) each sample points would have the same
    "weight" in the dating procedure; this should be useful for nearby volcanic eruptions, where the high concentration
    of sulphate could potentially mask the seasonal pattern, but as highlighted by bin plot and PCA, MSA maxima do
    not exactly match the other ions. For $NO_3^-$, its maxima can also be shifted by nearby volcanic eruptions due to the
    high acidity from $H_2SO_4$ (Jiang et al., 2019; Röthlisberger et al., 2000, 2002). For these reasons, this method was
    found not reliable in the lower section of the core as shown in Figure S3, where the different stratigraphies are
    compared.

3) Cation stratigraphies: although $Na^+$ and $Mg^{2+}$ stratigraphies were successfully used in the dating of ice cores
    (Herron and Langway, 1979; Winski et al., 2019) for the GV7 site they showed a less pronounced seasonal pattern
    both in the upper (as seen in the bin plots in Figure 4) and lower portions compared to the $nssSO_4^{2-}$ profile, as in
    Figure S4. Therefore, identification of winter maxima in their concentration profiles is not univocal, making
    impossible the accurate core dating.

4) As already seen in the first section of the core from the preliminary analysis of the different ions, the dating of the
    GV7 ice core using the $nssSO_4^{2-}$ concentration vs. depth profile without any further data manipulation is possible
    and the bin plot analysis suggests that this is, indeed, the best approach for dating the rest of the core.

The dating of the core was therefore carried out with a combination of annual layer counting and the identification of
volcanic signatures in the core. The know past volcanic eruptions found in other ice cores (Sigl et al., 2013, 2015, 2016;
Zielinski et al., 1996) as well as the tephra layer (Narcisi et al., 2001, 2012; Narcisi and Petit, 2021) and their assigned date
is reported in Table 1. Regardless of the exact date, due to the way the dating was finalized, the layers characterized by a
rising in the $nssSO4^{2-}$ concentration coinciding to a given eruption was assigned to the 1st of January in order to be
consistent with the dating of the seasonal maxima. The complete stratigraphy of the $nssSO_4^{2-}$ is reported in Figure S5 and the
finalized dating in Figure 5. When it comes to annual layer counting, a rigorous evaluation of the uncertainty of the dating is
difficult to accomplish and it is usually estimated based on the algorithm used to identify each annual layer (Sigl et al., 2016;
Winski et al., 2019) and/or by taking into account the uncertainty on the dating of different ice cores used as reference



(Winski et al., 2019). In this work we estimated the uncertainty over the annual layer counting as the sum of the layer
uncertainties highlighted in the dating procedure, estimated to be 0.5±0.5 years (Ramussen et al., 2006). The uncertainty was
estimated between each one of the known volcanic signatures highlighted in the ice core, dated with an uncertainty of ±1
year from the recorded eruption due to the amount of time needed to reach Antarctica.
The same level of uncertainty was assigned to the missing sections of the core, where the number of years present was
estimated using the average year/depth ratio calculated in 10 years before and after the break. Uncertainty levels are reported
in Table 2; the relatively higher number of uncertain layers in the lower section of the core is due to missing ice that led to a
non-continuous profile and to the lower resolution of the core, where each year could be represented by as low as 3 sample
points. Major volcanic eruptions in this section of the core are also fewer and far between each other (see Table 1), and the
lack of temporal horizon to constrain the dating, brought higher degree of uncertainty in the dating itself.

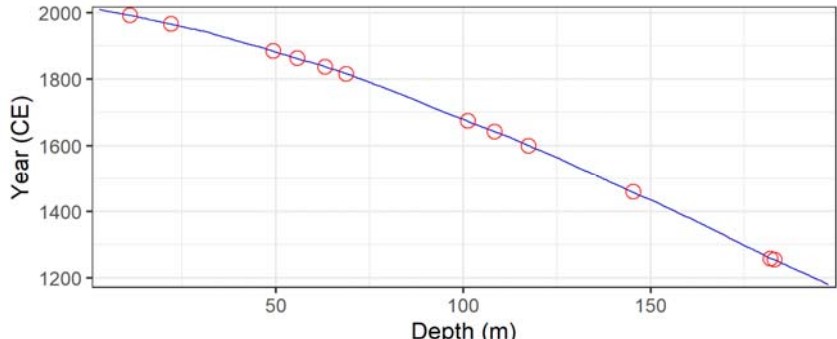


**Figure 5: age depth correlation for the GV7(B) ice core. Temporal horizons used as constraints in the dating procedure are**
**highlighted with red circles**

| Volcano | Depth (m) | Historical date (start) Year (CE) | Assigned date Year (CE) |
|---|---|---|---|
| **Pinatubo** | 11.10 | 1991 | 1992 |
| **Agung** | 22.12 | 1963 | 1965 |
| **Krakatoa** | 49.35 | 1883 | 1884 |
| **Makian** | 55.75 | 1861 | 1863 |
| **Cosiguina** | 63.27 | 1835 | 1837 |
| **Tambora** | 68.75 | 1815 | 1816 |
| **Gamkonora** | 101.25 | 1673 | 1675 |
| **Parker Peak** | 108.39 | 1641 | 1642 |
| **Huaynaputina** | 117.48 | 1600 | 1600 |
| **Reclus?** | 145.41 | 1460 | 1460 |





| | | | |
|---|---|---|---|
| **Samalas** | 181.86 | 1257 | 1258 |
| **Tephra Layer** | 183.07 | 1253 | 1254 |

**Table 1: known past volcanic eruptions used in the dating of the core**

| GV7 section (m) | Number of Annual Layers | | Duration (yrs) | Counting Error | |
|---|---|---|---|---|---|
| | Certain | Uncertain | | Absolute (yrs) | Percentage |
| 3.00 – 11.10 | 17 | 0 | 17 | - | - |
| 11.10 - 22-12 | 26 | 1 | 27 | 0.5 | 1.9 |
| 22.12 – 49.35 | 76 | 5 | 81 | 2.5 | 3.1 |
| 49.35 – 55.75 | 21 | 1 | 22 | 0.5 | 2.3 |
| 55.75 – 63.27 | 24 | 3 | 27 | 1.5 | 5.6 |
| 63.27 – 68.75 | 18 | 3 | 21 | 1.5 | 7.1 |
| 68.75 – 101.25 | 136 | 5 | 141 | 2.5 | 1.7 |
| 101.25 – 108.39 | 31 | 1 | 32 | 0.5 | 1.6 |
| 108.39 – 117.48 | 37 | 4 | 41 | 2 | 4.9 |
| 117.48 – 145.41 | 133 | 9 | 142 | 4.5 | 3.2 |
| 145.41 – 181.86 | 188 | 12 | 200 | 6 | 3.0 |
| 181.86 –183.07 | 4 | 0 | 4 | - | - |
| 183.07 – 197.00 | 69 | 6 | 75 | 3 | 4.0 |

**Table 2: Uncertainty levels over the GV7 ice core dating**

**3.3 Mean Snow Accumulation Rate evaluation**

Once the dating was finalized, for each year the amount of annual snow accumulated on the site was calculated in millimeter of water equivalent per year (mm w.e. yr-1) by multiplying the length of the core representative of any given year by the density of the core itself. The density (in g cm$^{-3}$, see Figure S4) was evaluated by weighting each section of the core.

Caiazzo et al. (2017) dating of a snow-pit by chemistry seasonal signal reports a mean accumulation rate of 242 mm w.e. yr$^{-1}$ for the period 2008-2013 CE with a Standard Deviation (SD) of 71 mm w.e. yr$^{-1}$, value very close to the estimation of the accumulation of the previous 50 years made by Magand et al., (2004) using atomic bomb horizon marker (241 ± 13 mm w.e. yr$^{-1}$ from 1965 to 2000 CE). Frezzotti et al. (2007) report an accumulation of 252 mm w.e. yr$^{-1}$ for the period 2001-2004 CE with a SD of 104 mm w.e. yr$^{-1}$ using snow stakes farm measurements and a mean accumulation of 237 mm w.e. yr$^{-1}$ for the



period 1854-2004 CE using the seasonal variation in nssSO$_4^{2-}$ concentrations, coupled with the identification of atomic bomb
markers and nssSO$_4^{2-}$ spikes from the most important past volcanic event.
The mean GV7(B) accumulation for periods 1965-2000 CE and 1854-2004 CE (242 with SD of 57 mm w.e. yr$^{-1}$ and 233
with SD of 64 mm w.e. yr$^{-1}$, respectively) confirm the ones found in the snow pits, stake measurements and shallow ice core
previously analyzed and covering different period during the last 150 years. On the other hand, the mean snow accumulation
calculated from the totality of the 195 m, representative for a time period ranging between and 1179 and 2009 CE, is 205
mm w.e. yr$^{-1}$ (SD of 63 mm w.e. yr$^{-1}$) lower than the one previously measured for the last century.
The comparison between GV7(B) and the ITASE record (Figure 6) highlights a similar trend especially in the period ranging
from 1900 to 2001 CE where the linear correlation between the two cores is high and significant (R=0.42, p<0.0001). On the
other hand, if we consider also part of the 18th century, the correlation decreases (R=0.3 p<0.0002) due to few
inconsistencies apparent between 1880 and 1850 CE probably due to the dating of the two cores, based on different spatial
sampling.
In order to remove the possible noise due to spatial variability (Frezzotti et al., 2007) and reduce the error connected to the
underestimating (overestimating) of the amount of yearly snow accumulated by misinterpreting the summer maxima in the
nssSO$_4^{2-}$ profile, a stacked record was obtained by combination of trench (2013-2008 CE), GV7B core (2009-1079 CE),
stake measurements (2003-2001) and ITASE core (2001-1849 CE). The new stacked record (Figure 6) can give valuable
information on snow accumulation trend in the Antarctic region through comparison with other ice cores drilled in the same
sector.  In the East Antarctic region, facing the Southern Indian Ocean, only three ice core records of snow accumulation
cover a period longer than three centuries: GV7 stacked (2013-1179 CE, this paper), Law Dome (2012 CE- 22 BCE, Roberts
et al., 2015) and Talos Dome (2010 -1217 CE; Stenni et al., 2001 Thomas et al., 2017). Other cores (D66, GV5, GV2, HN)
have been drilled but their records cover less than 300 years (Frezzotti et al., 2013; Thomas et al., 2017). Precipitations over
the GV7 area are related to storms coming from the Southern Indian Ocean (Caiazzo et al., 2017) as for Law Dome, whereas
the precipitation at Talos Dome coming only for 50% from Southern Indian Ocean and the remaining from the Ross Sea
(Sodemann and Stohl, 2009; Scarchilli et al. 2011). Law Dome (DSS) is a site close to the Southern Ocean (100 km from the
shoreline) at about 1400 m of elevation with a long term of accumulation of 740 mm w.e. yr$^{-1}$ (van Ommen et al., 2004),
about 1900 km west of GV7. Whereas Talos Dome is located at 2316 m and 250 km southern inland of GV7, with a long-
term accumulation of 80 mm w.e. yr-1 (Stenni et al., 2001). Roberts et al. (2015) pointed out that the two thousand years (22
BCE to 2012 CE) records at Law Dome shows no long-term trend in snow accumulation rates, however several anomalous
periods of accumulation exist in the record, most notably the periods of 380–442 CE, 727–783 CE and 1970–2009 CE (high
accumulation) and 663–704 CE, 933– 975 CE and 1429–1468 CE (low accumulation). Law Dome accumulation variability
is associated with both ENSO and IPO (Roberts et al., 2015; Vance et al., 2015), which influence the meridional component
of the large-scale circulation (van Ommen and Morgan, 2010; Roberts et al., 2015; Vance et al., 2015). For Talos Dome,
Stenni et al., (2001) pointed out for the period 1996-1217 CE a decrease during part of the Little Ice Age followed by an
increment of about 11% in accumulation during the 20th century.





The comparative analysis of the last 800 years of these three records shows a significant trend in accumulation record at
GV7 and Law Dome (Table 3), with a high increase in accumulated snow in the former and a slight increase in the latter (47
and 20 mm w.e., ~23% and ~2% of the mean accumulation over 800 years, respectively). On the other hand, no significant
trend at Talos Dome is apparent (Table 3).
Our analysis of variability at multi-centennial scale shows for GV7 a decrease of the accumulation rate from the beginning
of the record (1200 CE) to the middle of the 14th century. An analogous decrease has been already observed at Law Dome
(Roberts et al., 2015) and at Talos Dome (Stenni et al. 2001) (Figure 7). An increase in accumulation up to now is present at
GV7 and Talos Dome starting around middle 18th century (Table 3), whereas at Law Dome the data shows an increase a
century later from middle of 19th century. Decadal-scale snow accumulation anomalies were found at Law Dome to be
relatively common (74 events in the 2035-year record; Roberts et al., 2015). The previous study regarding the Talos Dome –
GV7 area, pointed out a century-scale variability with slight increase (of a few percent) in accumulation rates over the last
200 years, in particular since the 1960s, compared with the period 1816– 1965 CE (Frezzotti et al., 2007, 2013). At GV7 the
observed increase in accumulation during the last 250 years is greater than the observed range for the previous 600 years
(Figure 7). Frezzotti et al., 2013 analyzed 67 records from the entire continent over the last 800 yr to assess the temporal
variability of accumulation rates. The temporal and spatial variability of the records over the previous 800 yr indicates that
snow accumulation changes over most of Antarctica are statistically negligible and do not exhibit an overall long clear trend.
However, a clear increase in accumulation of more than 10% has observed in coastal and slope regions, as also this record
shows for GV7 site. Thomas et al., 2017 reveals that snow accumulation for the total Antarctica has increased since 1800
AD, where the annual snow accumulation during the most recent decade (2001–2010) is higher than the annual average at
the start of the 19th century. The Antarctic Peninsula is the only region where both the most recent 50- and 100- year trends
are greater than of the observed range for the past 300 years.




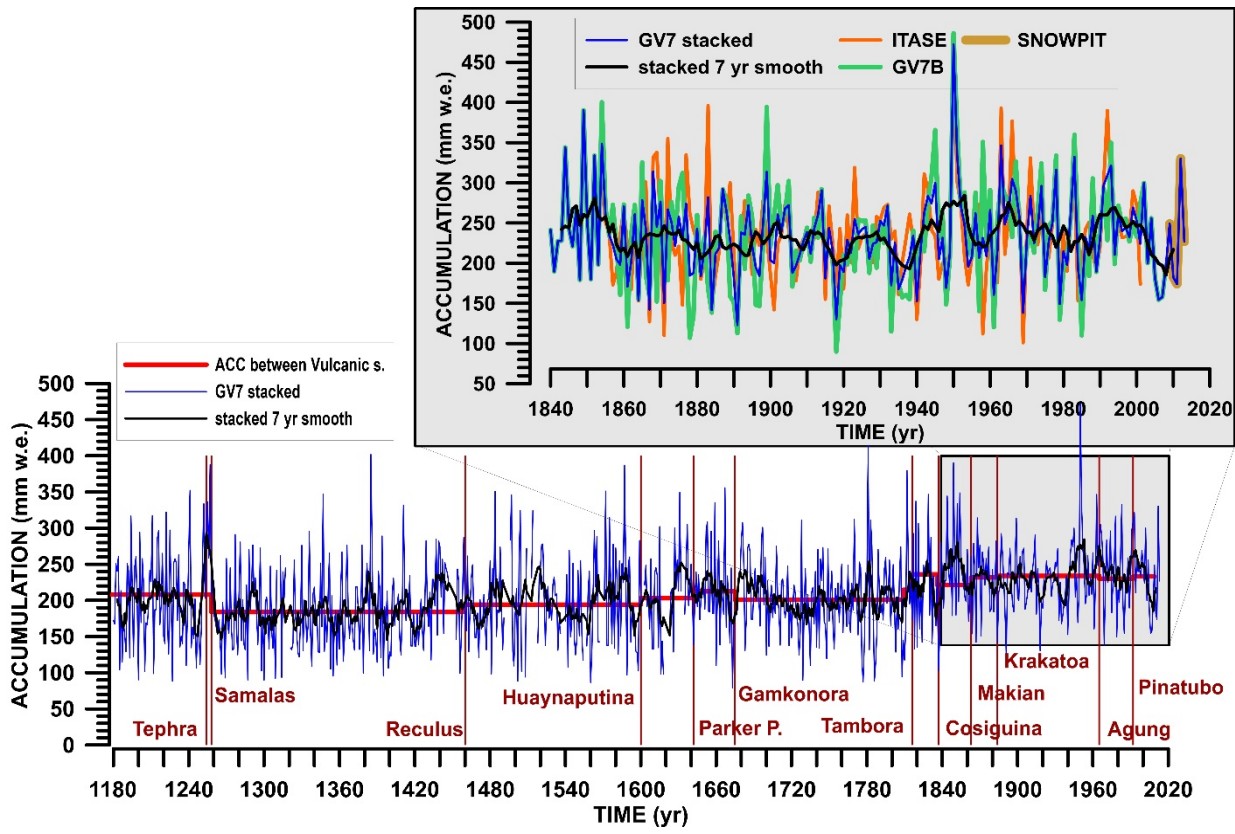

**Figure 6: a) Time series from 1840 to 2020 CE of the GV7 snowpit trench (2013-2008 CE, gold line); the ITASE core (2001-1849 CE, orange line) and the GV7 B core (2009-1179 CE, green line). Blue and black lines highlight the stacked record, obtained coupling the snowpit, ITASE and GV7 B core, and its smoothing at 7 years, respectively. b) GV7 stacked complete time record (1179-2013 CE) with seven years smoothing average (black line). Red vertical bars highlight volcanic eruption horizon and red line shows average accumulation between different volcanic events.**





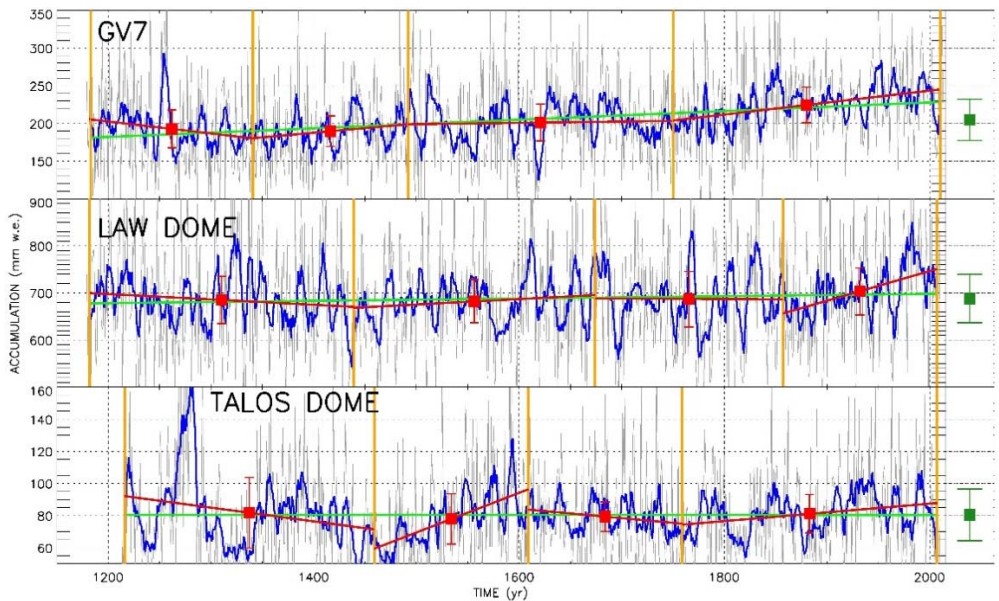

403

**Figure 7: a) GV7 stacked record (gray line) with its seven years smoothing average (blue line); green line represents trend for the 1179-2013 CE record. Yellow vertical bars show breaking points (1183 CE, 1341 CE, 1492 CE, 1750 CE and 2010 CE) calculated following Tomé and Miranda (2004). Red lines and filled squares show partial trends and mean accumulation (with standard deviation error bars) for each sub-period defined by breaking points. Green filled square with error bar highlights mean accumulation at the site and its standard deviation, respectively, for the whole period (1179-2013 CE). b) Same as A but for Law Dome ice core for the period 1179-2013 CE with breaking points at 1182 CE, 1439 CE, 1674 CE, 1857 CE and 2007 CE. c) Same as A but for Talos Dome ice core for the period 1216-2010 CE with breaking points at 1216 CE, 1459 CE, 1609 CE, 1759 CE and 2007 CE.**





412

| | 1183-2010 | 1183-1341 | 1341-1492 | 1492-1750 | 1750-2010 |
|---|---|---|---|---|---|
| **GV7** | Tr=+0.6 (p < 0.001) M=205 SD=27 | Tr=-1.6 (p < 0.001) M=193 SD=25 | Tr=1.2 (p < 0.001) M=190 SD=20 | Tr=0.2 (No sign.) M=201 SD=24 | Tr=1.6 (p < 0.001) M=224 SD=24 |
| **LAW DOME** | 1183-2007 Tr=-0.3 (p < 0.001) M=688 SD=52 | 1183-1439 Tr=-1.1 (p < 0.01) M=685 SD=50 | 1439-1674 Tr=1.1 (p < 0.01) M=682 SD=46 | 1674-1857 Tr=-0.1 (No sign) M=687 SD=59 | 1857-2007 Tr=6.3 (p < 0.001) M=704 SD=50 |
| **TALOS DOME** | 1216 -1996 No trend M=80 SD=16 | 1217-1459 Tr=-0.9 (p < 0.001) M=82 SD=22 | 1459-1609 Tr=2.5 (p < 0.001) M=78 SD=16 | 1609-1759 Tr=-0.6 (p < 0.001) M=79 SD=9 | 1759-2007 Tr=0.5 (p < 0.001) M=81 SD=12 |

413

**Table 3: Values of trend with significance (Tr, mm w.e./decade), mean accumulation and its standard deviation (M and Sd, respectively; mm w.e. yr-1) for GV7 stacked, Law Dome and Talos Dome smoothed with a 7-year running average, within different period defined by breaking points calculated following Tomé and Miranda (2004).**

## 4 Conclusion

In this work, we used the chemical stratigraphies obtained from the analysis of about 3500 discrete samples from the GV7(B) ice core to accurately date the core with a sub-annual resolution. $\delta^{18}O$ high resolution record was compared to $nssSO_4^{2-}$ profile showing negligible discrepancies. The two records were used to achieve a reliable dating of the uppermost section of the core (approx. 40m, covering the time period between 2009 and 1920 CE).

For the deeper section of the core, different strategies were tested and compared, namely single-parameter and multi-parametric approaches by considering seasonal markers to accomplish an annual layer counting. Upon these tests, $nssSO_4^{2-}$ profile was chosen for the dating of the core because of its clearer and better-preserved seasonal pattern all along the ice core, even at higher depth, where the temporal resolution becomes lower due to the thinning of the ice layers. An accurate annual layer counting was applied, and the volcanic signatures identified in the GV7 ice core were used as temporal horizons





and tie points in the dating procedure. In this way, an accurate dating of the core with a sub-annual resolution for the
uppermost 197 m was obtained.. Unfortunately, beyond the depth of 197 m, the ice core was strongly damaged and thus
heavily contaminated from the drilling fluid also in the inner part.
The time period covered by this uppermost 197 m of the core resulted to be 1179 - 2009 CE. In this period, an average
annual snow accumulation rate of 205 mm w.e. was calculated. Such value was compared with already available records
from the same site and different cores drilled in the same region (Law Dome and Talos Dome). Similar accumulation rate
was found when comparing it with another core drilled on the same site as part of the ITASE drilling campaign, with
particularly good agreement during the last 40 years. When considering the general trend of the accumulation throughout the
years, an increase was found since middle 18$^{th}$ century covered by the GV7(B) core. Such increasing trend has been
observed also at other slope coastal sites. Although the data here presented only cover the last 830 years, the number of cores
that cover the same time period is still scarce, therefore the present study could significantly contribute to the long-term
assessment of the surface mass balance in this area.

## 439 5 Acknowledgment

This research was financially supported by the MIUR (Italian Ministry of University and Research) - PNRA (Italian
Antarctic Research Programme) through the IPICS-2kyr-It project (International Partnership for Ice Core Science,
reconstructing the climate variability for the last 2kyr, the Italian contribution). The IPICS-2kyr-It project is carried out in
cooperation with KOPRI (Korea Polar Research Institute, grant No. PE21100).
The fellowship personnel involved in the analysis of the GV7 core, and the laboratory equipment was partially funded by
Fondazione Cassa di Risparmio di Firenze, MIUR-PNRA IPICS-2kyr-Italia (PNRA 2009/A2.09) MIUR-PNRA "BE-OI"
(PNRA16 00124) and "3D" (PNRA16 00212) projects and MIUR-PRIN 2017 "AMICO".

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
