# Peer review of "Dating of the GV7 East Antarctic ice core by high resolution chemical records and focus on the accumulation rate variability in the last millennium"

_Climate of the Past, 2021_

## Referee Comment (RC1)

**"Dating of an East Antarctic ice core (GV7) by high resolution chemical stratigraphies"**
**Nardin et al.** (2021)

Climate of the Past

5   Referee: D Emanuelsson

**Summary**

The age scale for a new ice core GV7, Oates Land, is presented.

The paper needs major revisions. There is no clear display material from which the reader can gain confidence in the age scale. This can be addressed by showing a figure with nssSO4 versus

10   time and indicate volcanic events that stand out from the background any by updating Table 1.

The novel aspects of the core, deepest and oldest core in the Oates region, the trace element records can be highlighted to increase the impact of this study. The writing needs to be improved and checked for clarity.

**Comments to the authors**

15   This is interesting work with some unique aspects. I hope the number of comments doesn't discourage you. Keep at it and this study will provide important records that can be used in future studies.

**Major comments:**

Be clearer about the method you use. You use nssSO4 peaks as your main indicator for assigning annual-layer counts, and you have $\delta^{18}O$ as additional support. Then this method is evaluated by

20   checking the dates against the volcanic events horizons. If they do not agree, you will have to check if your method is wrong, perhaps another chemical works better.

- Your pick at 4.67m (2005) could be questionable. As it is on a shoulder of the isotopes and the nssSO4 is low. Close to the surface, it is easy to overcount using the isotopes, as diffusion has not smoothed the signal as it has for a deeper section. Are there any other

25      impurities to look at? It almost looks like the trace elements could be helpful for the annual-layer counts. Check the figure that I merged below. (and the trace elements would maybe indicate that this is a pick?).

If you remove this pick, the years that you have marked with text around the Pinatubo eruption will be 1993 and 1992. Then the rise in the trace elements does not occur before the eruption.

30  The depths that you report for the trace element rise seem wrong (11.0 and 12.5 m). If you correct them (11.6 to 12.2 m) the trace element increase will not occur before the eruption. Don't eyeball this, use the criteria that you used in your earlier paper (2std or 3std above background), to find the exact depth/timing for these events.

Merge figures 2 and 3. Align b (now figure 3) with depth axis in the top panel of a (similar to
35  like I've done below). How does it look for Agung and other volcanic events? Or even better if there are trace elements for all the depths add two panels with trace elements (b, d). If there are trace elements for the core. It would be interesting to look at this closer and it could increase the impact of the paper.

[Figure]

40

**Fig. R1**

- How do the gaps in the gaps affect the accuracy of the dating? Add a supplementary table that provides % missing for each record.
- For uncertainty estimates, I prefer a method that compares ice core ages at volcanic event
45      horizons.

- Does the accumulation record have a correction for ice layer thinning? Please provide details. A long-term trend might be introduced by not making this correction.

50 **Specific comments:**

*Title*
Change tile. 'stratigraphies' typically is used when you refer to physical properties in a core, not chemical.

*Abstract*

55 L21. Stratigraphic dating, you could say this, but you can be misunderstood. The reader might think that you use layering in the ice from say radar scans not chemistry for dating. I suggest that you change this throughout the document.

L21. Say just 'drill site'.

L21. Provide the region where it was drilled Oates Land, East Antarctica.

60 L23. Say 'and water stable isotopes ($\delta^{18}$O)..' instead, or

L23. Rewrite. …sea salt ions contain a clear seasonal cycle and was therefore used for the annual-layer counting.'

L25. Delete 'correlation' say 'age-depth relationship' instead.

L25. '1179-2009' change the type of dash you use.

65 L27-28. 'A small, yet consistent, rise in accumulation rate was found for the last 830 years since the middle of the 18th century.' Is it significant? If not, it is not a trend. Provide trend and p-value.

Middle 18th century, approximately 1750, now it is 2021, that is about 370 yrs, not 830 years.

*Introduction*
70

L47. PNRA explain acronyms and explain them at their first occurrence (IPICS).

L58. Mention the work that Winstup et al. has done for constraining ice core chronologies (Winstrup et al. 2012, 2017)

L77. A too general statement, is it true for all Antarctic cores, or for GV7? If GV7 it should be reported in the result section.

L91 $\delta^{18}O$

L92 You mean that you linearly interpolate for the dates between annual markers? You can say that, but it should be in the method section not here.

L93 Just report on the methods that was successful here.

*Materials and methods*

- Why wasn't $\delta^{18}O$ analyzed throughout the core at high resolution? It would not be affected by contamination like the ions, so you could get a record with fewer gaps.

L98 insert a space, 1700 m.

L104 'Estimates of snow accumulation has been calculated from GPR layers from the 2001-2002 ITASE…'

L111 '(ranging in length between 5 to 50 m)'

L112 'The 250 m deep core, GV7(B) is used in this study. The ice core (Fig. 1) was retrieved using an electromechanical drill (Eclipse Ice drill Instrument).'

L119. Do you mean that you kept a 4 m of fluid measured from the drill bit to have the drill submerged but not more? The drill fluid surface was not kept at 80m by adding more fluid. This practice could perhaps be better when drilling a brittle core.

L123 be concise '…only the upper 194 m were analyzed.' Should suffice.

L128 and 133. Say cubes instead. The 4x4 cm core cubes for…

L143 and L144 Change to 'cations' and 'anions', remove the mentioning of the pump to be brief.

L147. 'while the analytical performance of the methods was tested and described in Nardin et al., (2020).' I do not think you can claim that you did this in your paper. You showed a high-resolution record, but no specific tests of the performance.

L156. Remove the dash and add punctuation.

100   L174 and L175 Do you use these ratios in the manuscript?

L179. Rewrite this section …was used to identify volcanic.. end sentence after 'ice cores'.

L184. Delete 'where an in-depth discussion….' While this is true, it is not relevant here. Mention it in another section.

L200. Due 'to' possible noise…

105   *Result and discussion*

L211. You use nssSO4 and $\delta^{18}O$ together to make a judgment for the annual marker, correct? Here it sounds like you make to age scales one using $\delta^{18}O$ and one using nssSO4.

L215. You cannot expect the peaks of $\delta^{18}O$ and nssSO4 to always align. Especially not for
110   deeper sections when diffusion has acted on the isotopic signal. It is not caused by the different depth resolution of the records. Remove the last two sentences of this paragraph.

L256. This sentence is way too long. And the end can be misunderstood. It is enough to say. '.., suggesting a lack of seasonality.'

L271. The punctuation is missing.

115   L274. Change Factor 1 and 2 to PC1 and PC2 (principal components).

L275. The signs of the PCs are arbitrary.

L268. Delete the PCA analysis. I do not think it adds any value. Are you using it for dating?

L278 In general, you do not need to report all approaches that you try that turn out to not be fruitful, this makes the paper long and hard to read.

120   L294 Should be 'unequivocal'.

L316. Rewrite this sentence. Suggestions, 'there are fewer major eruptions in this section of the core and as they are further apart the uncertainty becomes larger.

L328-329. The first sentences here should be in the method sections. Add a SMB section to the methods and provide more detail about thinning correction too. For example, Thomas et al. (2017) have a method section describing this correction.  Present your results first here and then discuss with others' work.

L332. You use several symbols and abbreviations for standard deviation, stick to just one.

L344. Rewrite for clarity. The mixture of trends and correlations in this paragraph is confusing.

L349. What do you mean here by 'misinterpretation'? If you show that your annual layer counting method works. That is nssSO4 peaks line up with known volcanic horizons. Then you cannot at the same time claim that you have done a misinterpretation.

L350. Switch the order of the years around.

L417. Change to 'Conclusions'.

L430. Change to 'The GV7 chronology covers the 1179-2009 CE period. The average annual snow accumulation for this period is 205 mm w.e.'

Add a data availability section.

***Figures***
Figure 1.

Zoom in on the region of interest. Show the whole Antarctic map as an insert. The figure will look better if you add topography (example from bedmap2).

L115 Change to '…1950 m a.s.l.) drill site…'.

Figure 2.

Pinatubo is marked but not Agung.

Have you removed single value peaks, contamination? Using the code you described in (Nardin et al. 2020).

Show lines for each annual pick. Zoom in a bit, so it doesn't become too cluttery.

Change the coloring on the y-axes so they correspond to what is shown.

I prefer the figure you have in the supplementary material (Figs. S3 and S4) over this, as $\delta^{18}O$ is only available for the top part of the core.

150    Figure 4.

'Concentration bins have a three-month seasonal width..'. Tell the reader which intervals you use here, JFM, AMJ,…

'Upper concentration limits and bin sizes were chosen to keep between each ion's plot the same proportions in order to facilitate the interpretation of the data.' Do you need to say this, or do you

155    mean that you have cut out some data with high concentrations?

As nssSO4 is used as the primary chemical for pick annual layers, it will because of this choice show more seasonality. I am not questioning your choice, I'm merely pointing this out.

Figure 5. Change to: 'Age-depth relationship…'

Figure 6.

160    Is there a periodicity in the data, is there an increase after a volcanic event?
To check this you can make a plot with all the events and put each event at time zero, so you can plot them together.

Change axis text from 'TIME' to Time and ACCUMULATION to Accumulation.

Add letters to the figure indicating subplots (a, b).

165    Start with the start year, i.e. 1849-2001 CE.

Indicate 'Tephra' with another color in the figure as it is not a well-established event.

Figure 7. Say 'Accumulation records used for the core stack (gray lines), the stacked accumulation record (blue line)'

Add letters (a, b, c) to the figure.

170    L410. 'a' should not be capitalized.

Which specific ITASE core was this? There are many.

Table 1.

'Pinatubo/Cerro Hudson'

Table captions usually go on top of the table.

175    Is it the start or end depth that you are providing? Provide an end and start date, which is more exact. You can get this from Figure S5. Pinatubo 1991.5 (start) 1993.5 (end) (e.g. from the WAIS core, (Sigl et al. 2013)).

Table 3.

Rewrite caption. 'its'

180    If you smooth the record before the analysis that would affect the standard deviation.

Change to 'not significant' instead of 'No sign.' Or 'No trend'.

L414. Rewrite this sentence. Suggestion: 'the mean (M) and standard deviation (SD) of the accumulation record..'

It is also confusing that you call the stacked record used here the same name as the other single

185    core.

*Supplementary material*

Table S1 and Figure S1. Provide the same information. Stick with just the biplot.

Table S1 caption. The marked loading are >.6? PC1 explain 0.40 and PC2 0.184. (= 58.6%

190    explained together)?

Do you mean that it has been varimax rotated?

You need punctuation at the end of each caption.

Figure S3. Move one of these figures to the main text. Remove the normalized record, if it doesn't add anything. Check with a seasonal plot if you are not sure (like Fig. 4).

195    Remove the parentheses pair after used after 'layering (…)'.

Typo should be 'orange line'.

Figure S4. 'for a shallow (a) and a deep (b) section of the core'.

'Despite showing a similar…' remove this sentence. I would avoid having this discussion in the figure caption.

200 Figure S5.

Show the whole record. Do not start at 25 m.

Change to nssSO4 versus time. What you show is the same as in Nardin et al. 2020 otherwise.

Add volcanic events. Like you did nicely in Nardin et al. 2020, but now do it versus time. Add the background +2std and 3std levels. This way you can evaluate the timing of the volcanic

205 peaks and the reader can gain confidence in the age scale. It is a large figure, but important, I would consider moving it into the main text.

Increase the line thickness of the nssSO4 line and the box around the figure and make the text bold. This will improve the figure quality.

'with grey dashed and red lines, respectively.

210 'Other breakages are not highlighted' remove this sentence.

**References**

Nardin R, Amore A, Becagli S, et al (2020) Volcanic Fluxes Over the Last Millennium as Recorded in the Gv7 Ice Core (Northern Victoria Land, Antarctica). Geosci.  10

215 Nardin R, Severi M, Amore A, et al (2021) Dating of an East Antarctic ice core (GV7) by high resolution chemical stratigraphies. Clim Past Discuss 2021:1–27. https://doi.org/10.5194/cp-2021-44

Sigl M, McConnell JR, Layman L, et al (2013) A new bipolar ice core record of volcanism from WAIS Divide and NEEM and implications for climate forcing of the last 2000 years. J Geophys Res Atmos 118:1151–1169. https://doi.org/10.1029/2012JD018603

220 Thomas ER, van Wessem JM, Roberts J, et al (2017) Regional Antarctic snow accumulation over the past 1000 years. Clim Past 13:1491–1513. https://doi.org/10.5194/cp-13-1491-2017

Winstrup M, Svensson  a. M, Rasmussen SO, et al (2012) An automated approach for annual layer counting in ice cores. Clim Past 8:1881–1895. https://doi.org/10.5194/cp-8-1881-2012

Winstrup M, Vallelonga PT, Kjær H a., et al (2017) A 2700-year timescale and accumulation
225 reconstruction for Roosevelt Island, West Antarctica. Clim Past Discuss submitted

---

## Author Comment (AC1)

**We thank the reviewer for his careful and constructive comments and inputs. Here below are reported our answers to the questions raised by the referee starting with "Reply".**

**"Dating of an East Antarctic ice core (GV7) by high resolution chemical stratigraphies" Nardin et al. (2021)**

**Climate of the Past**

**Referee: D Emanuelsson**

**Summary**

The age scale for a new ice core GV7, Oates Land, is presented.

The paper needs major revisions. There is no clear display material from which the reader can gain confidence in the age scale. This can be addressed by showing a figure with nssSO4 versus time and indicate volcanic events that stand out from the background any by updating Table 1.

The novel aspects of the core, deepest and oldest core in the Oates region, the trace element records can be highlighted to increase the impact of this study. The writing needs to be improved and checked for clarity.

**Comments to the authors**

This is interesting work with some unique aspects. I hope the number of comments doesn't discourage you. Keep at it and this study will provide important records that can be used in future studies.

**Major comments:**

- Be clearer about the method you use. You use nssSO4 peaks as your main indicator for assigning annual-layer counts, and you have δ18O as additional support. Then this method is evaluated by checking the dates against the volcanic events horizons. If they do not agree, you will have to check if your method is wrong, perhaps another chemical works better.

Reply: Thanks for the comment. We actually have worked as the Referee guessed: nssSO4 peaks were used for the annual layer counting and d18O record was used as a confirmation of the correct (or most reliable, anyway) assignment. We have indeed checked the possibility of using other seasonal chemical markers or a combination of them (as briefly described in the text) but their seasonal signals are generally less clear than nssSO4 and, mostly, they are not accountable on the long-term, being barely detectable in some depth intervals. Sections from Methods and Results will be rewritten accordingly.

- Your pick at 4.67m (2005) could be questionable. As it is on a shoulder of the isotopes and the nssSO4 is low. Close to the surface, it is easy to overcount using the isotopes, as diffusion has not smoothed the signal as it has for a deeper section. Are there any other impurities to look at? It almost looks like the trace elements could be helpful for the annual-layer counts. Check the figure that I merged below. (and the trace elements would maybe indicate that this is a pick?).

  If you remove this pick, the years that you have marked with text around the Pinatubo eruption will be 1993 and 1992. Then the rise in the trace elements does not occur before the eruption. The depths that you report for the trace element rise seem wrong (11.0 and 12.5 m). If you correct them (11.6 to 12.2 m) the trace element increase will not occur before the eruption. Don't eyeball this, use the criteria that you used in your earlier paper (2std or 3std above background), to find the exact depth/timing for these events.

Reply: Thank you for pointing this out. We had noticed that this peak was just a shoulder on both the records we showed in the paper, and therefore the date assignment could be questionable. Anyway, we decided to choose it as a summer maximum due to the following points:

1. We did consider other ions, as reported in the figure below. Sodium profile showed a clear minimum corresponding with the shoulder at 4.67 m. Although this cannot be considered as a conclusive evidence, it suggests that such a peak is likely to have occurred in summer rather than in winter.

2. Without considering the peak at 4.67 m depth as a seasonal maximum, we would obtain an unrealistic value of the accumulation rate between the previous and the following maximum. In fact, missing this year would yield a two-fold annual snow accumulation rate with respect to the average observed at the site.

As for the trace elements, we agree that removing the peak at 4.67 m depth would make more coeval the peaks of trace elements and nssSO4 corresponding to the Pinatubo eruption but we would be in favour of keeping the year assignment as it is since this one-year change would shift the age scale within the estimated uncertainty interval.

[Figure]

Figure REW1. nssSO4 (orange line) and Sodium (blue line) profiles between 3.5 m and 6 m below surface; y axis = concentration as ppb; x axis = depth as meters down from surface

- Merge figures 2 and 3. Align b (now figure 3) with depth axis in the top panel of a (similar to like I've done below). How does it look for Agung and other volcanic events? Or even better if there are trace elements for all the depths add two panels with trace elements (b, d). If there are trace elements for the core. It would be interesting to look at this closer and it could increase the impact of the paper.

Reply: Thank you for the comment. As suggested by the reviewer, we merged the two figures and aligned the three (six) records in order to make the whole dataset easier to read and to understand. Unfortunately, the trace element record below 20 m depth is not yet available and to date it is not possible to infer any information about a possible signature of the Agung eruption from trace metals.

Fig. R1

• How do the gaps in the gaps affect the accuracy of the dating? Add a supplementary table that provides % missing for each record.

Reply: Thank you for pointing this out. The missing ice certainly affects the accuracy of our dating because we have to rely on a continuous record in order to be able to date the core. For the missing section of the core we estimated the number of the missing years on the basis of the annual peak frequency in the time window spanning 20 years and centered on the missing ice section. We will add a section in Table 2 with the missing percentage of the core for each section between volcanic signatures.

• For uncertainty estimates, I prefer a method that compares ice core ages at volcanic event horizons.

Reply: Thank you for pointing this out, the uncertainty levels were re-evaluated. Table 2 will be revised according to your inputs and those from Reviewer #2 adding more detailed information.

• Does the accumulation record have a correction for ice layer thinning? Please provide details. A long-term trend might be introduced by not making this correction.

Reply: Thank you for pointing this out. Ice layer thinning was kept in consideration when evaluating snow accumulation rate on site. A constant ice thinning function was used to correct the data considering an ice thickness of 1530 m. This information will be added in the main text.

**Specific comments:**

**Title**

Change tile. 'stratigraphies' typically is used when you refer to physical properties in a core, not chemical.

Reply: we thank the reviewer for the suggestion. The term "records" will be used instead of "stratigraphies".

According to your advice for rephrasing and following on the suggestion by the Editor for better addressing the reader we would be prone for the following new title:

"Dating of the GV7 East Antarctic ice core by high resolution chemical records and focus on the accumulation rate variability in the last millennium"

**Abstract**

L21. Stratigraphic dating, you could say this, but you can be misunderstood. The reader might think that you use layering in the ice from say radar scans not chemistry for dating. I suggest that you change this throughout the document.

Reply: thank you for the comment. We'll revise the wording of the text in order to avoid misunderstandings.

L21. Say just 'drill site'.

Reply: it will be done.

L21. Provide the region where it was drilled Oates Land, East Antarctica.

Reply: it will be added in the text, where needed.

L23. Say 'and water stable isotopes (δ18O)..' instead, or

Reply: Thank you, it will be changed as suggested.

L23. Rewrite. …sea salt ions contain a clear seasonal cycle and was therefore used for the annual-layer counting.'

Reply: Thank you, it will be corrected

L25. Delete 'correlation' say 'age-depth relationship' instead.

Reply: Thank you, it will be changed

L25. '1179-2009' change the type of dash you use.

Reply: it will be changed

L27-28. 'A small, yet consistent, rise in accumulation rate was found for the last 830 years since the middle of the 18th century.' Is it significant? If not, it is not a trend. Provide trend and pvalue. Middle 18th century, approximately 1750, now it is 2021, that is about 370 yrs, not 830 years.

Reply: p value and trend for the investigated period will be added in the abstract. Table 3 will report the trend values of the accumulation rate between consecutive eruptions.

**Introduction**

L47. PNRA explain acronyms and explain them at their first occurrence (IPICS).

Reply: all the acronyms will be made explicit at their first occurrence

L58. Mention the work that Winstup et al. has done for constraining ice core chronologies (Winstrup et al. 2012, 2017)

Reply: Thank you for bringing this to our attention, they will be added

L77. A too general statement, is it true for all Antarctic cores, or for GV7? If GV7 it should be reported in the result section.

Reply: Thank you for the comment. This is generally true for a number of high accumulation sites where nitrate is preserved and was observed to show a seasonal pattern (Stenni et al. 2001 JGR; Wolff, 1995; Wagenbach, 1998 JGR). GV7 is a coastal-like, relatively high accumulation site, not showing the post-depositional processes usually affecting nitrate in inner plateau sites and it was worth trying nitrate as seasonal marker.

Some information will be added in order to clarify the statement, also complying with the remarks coming from Prof. Cole-Dai.

L91 δ18O

Reply: Thank you, it will be corrected

L92 You mean that you linearly interpolate for the dates between annual markers? You can say that, but it should be in the method section not here.

Reply: Thank you for your comment. The reviewer got it right: we have linearly interpolated between annual maxima. As suggested, we will move this information to the Methods section.

L93 Just report on the methods that was successful here.

Reply: Thank you, the sentence will be rephrased in order to stress the chosen dating approach.

**Materials and methods**

• Why wasn't $\delta^{18}O$ analyzed throughout the core at high resolution? It would not be affected by contamination like the ions, so you could get a record with fewer gaps.

Reply: Thank you for the comment. Unfortunately, high resolution d18O measurements are not yet available and cannot be added to this manuscript.

However, thanks to the manual decontamination of ice section that we have accomplished, the contamination was found only for a few samples. The main reason for gaps in the records is mainly due to breaks and missing core sections, rather than contaminated samples.

L98 insert a space, 1700 m.

Reply: Thank you, it will be corrected

L104 'Estimates of snow accumulation has been calculated from GPR layers from the 2001-2002 ITASE...'

Reply: Thank you, it will be changed

L111 '(ranging in length between 5 to 50 m)'

Reply: Thank you, it will be corrected

L112 'The 250 m deep core, GV7(B) is used in this study. The ice core (Fig. 1) was retrieved using an electromechanical drill (Eclipse Ice drill Instrument).'

Reply: Thank you, it will be corrected

L119. Do you mean that you kept a 4 m of fluid measured from the drill bit to have the drill submerged but not more? The drill fluid surface was not kept at 80m by adding more fluid. This practice could perhaps be better when drilling a brittle core.

Reply: thank you, the reviewer is right. A level of 4 m of fluid able to submerge the driller was chosen as the best compromise between drilling efficiency and quality of the core.

L123 be concise '…only the upper 194 m were analyzed.' Should suffice.

Reply: Thank you, it will be changed

L128 and 133. Say cubes instead. The 4x4 cm core cubes for…

Reply: I think the text was probably not clear enough, they're not cubes. We received ice 4x4x60 strips to be decontaminated. This information will be added to the text.

L143 and L144 Change to 'cations' and 'anions', remove the mentioning of the pump to be brief.

Reply: Thank you, the paragraph will be rewritten without Ion Chromatography technical details

L147. 'while the analytical performance of the methods was tested and described in Nardin et al., (2020).' I do not think you can claim that you did this in your paper. You showed a high resolution record, but no specific tests of the performance.

Reply: Thank you for pointing this out, it will be removed from the text

L156. Remove the dash and add punctuation.

Reply: Thank you, it will be done

L174 and L175 Do you use these ratios in the manuscript?

Reply: Thank you for your comment, these ratios were used in the paper.

L179. Rewrite this section …was used to identify volcanic.. end sentence after 'ice cores'.

Reply: Thank you, the sentence will be rewritten

L184. Delete 'where an in-depth discussion….' While this is true, it is not relevant here. Mention it in another section.

**Reply:** Thank you, it will be removed from this section.

L200. Due 'to' possible noise…

**Reply:** Thank you, it will be edited

**Result and discussion**

L211. You use nssSO4 and δ18O together to make a judgment for the annual marker, correct? Here it sounds like you make to age scales one using δ18O and one using nssSO4.

**Reply:** the reviewer is correct: the original sentence in the paper was misleading. As above said, nssSO4 record was selected to lead the counting of annual maxima and d18O was used to support in the detection of single years. The sentence will be rephrased to avoid misunderstandings.

L215. You cannot expect the peaks of δ18O and nssSO4 to always align. Especially not for deeper sections when diffusion has acted on the isotopic signal. It is not caused by the different depth resolution of the records. Remove the last two sentences of this paragraph.

**Reply:** Thank you for pointing this out, the referee is right. The sentences will be removed.

L256. This sentence is way too long. And the end can be misunderstood. It is enough to say. '.., suggesting a lack of seasonality.'

**Reply:** Thank you, the sentence will be split and the suggested change will be applied.

L271. The punctuation is missing.

**Reply:** Thank you, punctuation will be fixed.

L274. Change Factor 1 and 2 to PC1 and PC2 (principal components).

**Reply:** Thank you for pointing this out, PCA will be removed as suggested later

L275. The signs of the PCs are arbitrary.

**Reply:** Thank you for pointing this out, PCA will be removed as suggested later

L268. Delete the PCA analysis. I do not think it adds any value. Are you using it for dating?

**Reply:** Thank you for the comment. We agree with the Reviewer, it is not relevant for the purpose of the paper. It will be removed from the Supplementary information.

L278 In general, you do not need to report all approaches that you try that turn out to not be fruitful, this makes the paper long and hard to read.

Reply: Thank you for pointing this out. We are going to shorten this overview of dating approaches, but we think that such information could be useful for scientists involved in dating exercises, considering the site-to-site variability of chemical features.

L294 Should be 'unequivocal'.

Reply: Thank you, it will be corrected

L316. Rewrite this sentence. Suggestions, 'there are fewer major eruptions in this section of the core and as they are further apart the uncertainty becomes larger.

Reply: Thank you for the remark, the sentence will be changed as suggested

L328-329. The first sentences here should be in the method sections. Add a SMB section to the methods and provide more detail about thinning correction too. For example, Thomas et al. (2017) have a method section describing this correction. Present your results first here and then discuss with others' work.

Reply: Thank you for the comment, the sentences will be moved in a new Method section dealing with SMB.

L332. You use several symbols and abbreviations for standard deviation, stick to just one.

Reply: Thank you for the comment. The suggested changes will be applied.

L344. Rewrite for clarity. The mixture of trends and correlations in this paragraph is confusing.

Reply: Thank you for the comment. This paragraph will be rewritten in order to be clearer.

L349. What do you mean here by 'misinterpretation'? If you show that your annual layer counting method works. That is nssSO4 peaks line up with known volcanic horizons. Then you cannot at the same time claim that you have done a misinterpretation.

Reply:  the reviewer is right in noticing this possible contradiction. We agree with the reviewer that, once the dating is carried out, there is no space for "misinterpretation" but just uncertainty. The wording of paragraph will be checked and adjusted accordingly.

L350. Switch the order of the years around.

Reply:  thank you for the remark; the reviewer is right. We will shift the order of the years.

L417. Change to 'Conclusions'.

Reply: Thanks. The correction will be accomplished.

L430. Change to 'The GV7 chronology covers the 1179-2009 CE period. The average annual snow accumulation for this period is 205 mm w.e.'

Reply: Thanks. The correction will be accomplished.

**Add a data availability section.**

Reply: Thank you. This section will be added to the manuscript.

**Figures**

Figure 1.

Zoom in on the region of interest. Show the whole Antarctic map as an insert. The figure will look better if you add topography (example from bedmap2).

L115 Change to '…1950 m a.s.l.) drill site…'.

Reply: Thank you for pointing this out. The image was re-plotted using Matlab extension (bedmap2 and AMT) focusing on the region of interest.

The caption will be updated.

Figure 2.

Pinatubo is marked but not Agung.

Have you removed single value peaks, contamination? Using the code you described in (Nardin et al. 2020). Show lines for each annual pick. Zoom in a bit, so it doesn't become too cluttery. Change the coloring on the y-axes so they correspond to what is shown.

I prefer the figure you have in the supplementary material (Figs. S3 and S4) over this, as δ18O is only available for the top part of the core.

Reply: Thank you for your comment. Figure 2 will be updated according to the reviewer's suggestions to provide more information.

The reviewer is right about the removal of single value peaks likely due to contamination.

Figure 4.

'Concentration bins have a three-month seasonal width..'. Tell the reader which intervals you use here, JFM, AMJ,…

'Upper concentration limits and bin sizes were chosen to keep between each ion's plot the same proportions in order to facilitate the interpretation of the data.' Do you need to say this, or do you mean that you have cut out some data with high concentrations?

As nssSO4 is used as the primary chemical for pick annual layers, it will because of this choice show more seasonality. I am not questioning your choice, I'm merely pointing this out.

Reply: Thank you for pointing this out, the plot was edited as suggested. The bin interval (JFM, AMJ, JAS, OND) will be reported both in the figure and in the caption.

We actually do not need to say "Upper concentration limits and bin sizes were chosen to keep between each ion's plot the same proportions in order to facilitate the interpretation of the data" and the sentence will be removed.

Figure 5.

Change to: 'Age-depth relationship…'

Reply: Thank you for pointing this out, it will be edited

Figure 6.

Is there a periodicity in the data, is there an increase after a volcanic event?

To check this you can make a plot with all the events and put each event at time zero, so you can plot them together.

Change axis text from 'TIME' to Time and ACCUMULATION to Accumulation.

Add letters to the figure indicating subplots (a, b).

Start with the start year, i.e. 1849-2001 CE.

Indicate 'Tephra' with another color in the figure as it is not a well-established event.

Reply: Thank you for the hints. As regarding the figure, we will improve it taking into account all the reviewer's suggestions. About the study of periodicity of accumulation, also in relation with volcanic events, it is certainly something worth of further investigation, but we think that it is beyond the scope of the paper which is mainly focused on the production of a reliable age-scale for GV7 core.

Figure 7.

Say 'Accumulation records used for the core stack (gray lines), the stacked accumulation record (blue line)'

Add letters (a, b, c) to the figure.

'a' should not be capitalized.

Which specific ITASE core was this? There are many.

Reply: Thank you for the suggestions. They will be taken into account to revise the Figure.

Table 1.

'Pinatubo/Cerro Hudson'

Table captions usually go on top of the table.

Is it the start or end depth that you are providing? Provide an end and start date, which is more exact. You can get this from Figure S5. Pinatubo 1991.5 (start) 1993.5 (end) (e.g. from the WAIS core, (Sigl et al. 2013)).

Reply: Thank you for your comment. We will update the table being clearer about the start and the end of the signature but we would stick to the integer of the year. Unfortunately, our temporal resolution does not allow reporting accurately the beginning and end of each signature as accomplished by Sigl et al., 2013.

Table 3.

Rewrite caption. 'its'

If you smooth the record before the analysis that would affect the standard deviation.

Change to 'not significant' instead of 'No sign.' Or 'No trend'.

L414. Rewrite this sentence. Suggestion: 'the mean (M) and standard deviation (SD) of the accumulation record..'

It is also confusing that you call the stacked record used here the same name as the other single core.

Reply: Thank you for the comment. The reviewer is right: standard deviations will be removed from Table 3.

Moreover, the stacked record reported in Table 3 is now called "GV7 stacked".

**Supplementary material**

Table S1 and Figure S1. Provide the same information. Stick with just the biplot.

Table S1 caption. The marked loading are >.6? PC1 explain 0.40 and PC2 0.184. (= 58.6% explained together)?

Do you mean that it has been varimax rotated?

You need punctuation at the end of each caption.

Reply: Thank you for the comments. Table S1 will be removed and we'll keep the biplot. Punctuation will be fixed.

Figure S3. Move one of these figures to the main text. Remove the normalized record, if it doesn't add anything. Check with a seasonal plot if you are not sure (like Fig. 4).

Remove the parentheses pair after used after 'layering (…)'.

Typo should be 'orange line'.

Reply: Thank you for the accurate remarks. We'll correct the caption as suggested. About the location of Figure S3 in the manuscript we would like to keep it in the Supplementary Information since it is made of two enlarged views of chemical records in sections recording volcanic eruptions, spanning short depth intervals. Moreover, it includes chemical markers that were tested but then not used for annual layer counting.

Figure S4. 'for a shallow (a) and a deep (b) section of the core'.

'Despite showing a similar…' remove this sentence. I would avoid having this discussion in the figure caption.

Reply: Thank you for pointing this out, the caption will be edited.

Figure S5.

Show the whole record. Do not start at 25 m.

Change to nssSO4 versus time. What you show is the same as in Nardin et al. 2020 otherwise.

Add volcanic events. Like you did nicely in Nardin et al. 2020, but now do it versus time. Add the background + 2std and 3std levels. This way you can evaluate the timing of the volcanic peaks and the reader can gain confidence in the age scale. It is a large figure, but important, I would consider moving it into the main text.

Increase the line thickness of the nssSO4 line and the box around the figure and make the text bold. This will improve the figure quality.

'with grey dashed and red lines, respectively.

'Other breakages are not highlighted' remove this sentence.

Reply: Thank you for your comment, the plots will be edited as suggested and moved to the main text.

**References**

Nardin R, Amore A, Becagli S, et al (2020) Volcanic Fluxes Over the Last Millennium as Recorded in the Gv7 Ice Core (Northern Victoria Land, Antarctica). Geosci. 10

Nardin R, Severi M, Amore A, et al (2021) Dating of an East Antarctic ice core (GV7) by high resolution chemical stratigraphies. Clim Past Discuss 2021:1–27. https://doi.org/10.5194/cp-2021-44

Sigl M, McConnell JR, Layman L, et al (2013) A new bipolar ice core record of volcanism from WAIS Divide and NEEM and implications for climate forcing of the last 2000 years. J Geophys Res Atmos 118:1151–1169. https://doi.org/10.1029/2012JD018603

Thomas ER, van Wessem JM, Roberts J, et al (2017) Regional Antarctic snow accumulation over the past 1000 years. Clim Past 13:1491–1513. https://doi.org/10.5194/cp-13-1491-2017

Winstrup M, Svensson a. M, Rasmussen SO, et al (2012) An automated approach for annual layer counting in ice cores. Clim Past 8:1881–1895. https://doi.org/10.5194/cp-8-1881-2012

Winstrup M, Vallelonga PT, Kjær H a., et al (2017) A 2700-year timescale and accumulation reconstruction for Roosevelt Island, West Antarctica. Clim Past Discuss submitted

---

## Author Comment (AC2)

**We thank the reviewer for his careful and constructive comments and inputs. Here below are reported our answers to the questions raised by the referee starting with "Reply".**

**"Dating of an East Antarctic ice core (GV7) by high resolution chemical stratigraphies" Nardin et al. (2021)**

**Climate of the Past**

**Referee: Anders Svensson,**

The manuscript is concerned with a layer-counted dating of the uppermost 197 m of an East Antarctic ice core named GV7. The core is previously dated by identification of volcanic reference horizons, so the contribution of the manuscript is the presentation of the high-resolution chemical records and the application of those for layer counting in between the volcanic reference horizons. An interesting outcome is the identification of a slight increase in accumulation over the last centuries, when the GV7 record is stacked with other cores.

**Major comments**

> The MS needs to be reworked concerning the language. Sections 1 and 2 are fairly readable, but section 3 is not concisely written, it is full of syntax issues, and it is often hard to follow the argumentation mostly due to the language, I believe. Sorry to be this bold, but the writing is not up to scientific standards and needs to be reworked. I'm not going to make a detailed list of all of the minor issues in this review, as I think it is not the task of the reviewer, but I'm happy to read the manuscript again in a revised form.

Reply: Thank you for your comments. English language will be checked all through the text aiming to improve its readability, with particular attention to Section 3.

> Since the layer counting is one of the main contributions of this MS, it is important that is done correctly. I have an issue with the numbers stated in Table 2: It says that the layer counting follows the method of Rasmussen et al., 2006, where 'uncertain' years are counted as ½ ± ½ year. Now in table 2, say for the interval 145.41-181.86 m depth, you have identified 188 certain and 12 uncertain layers. According to the Rasmussen counting approach this should lead to an interval duration of:

> $188 + ½ \times 12 ± ½ \times 12$ years $= 194 ± 6$ years.

> In the table however, the duration of the interval is stated to be 200±6 years. How does this add up?

Reply: Thank you for the very insightful comment. The reviewer is right in stressing the discrepancy between what stated in the text (use of Rasmussen counting approach and evaluation of related uncertainty) and what reported in the Table. We thank the reviewer for spotting a mistake we made during the compilation of the Table. In fact, the number of uncertain layers was wrongly reported as the half of

what actually counted. We will revise Table 2 correcting the column "Uncertain Annual Layers" and, consequently the column "dating error Abs".

The '4-seasons' approach taken in Figure 4 is interesting, but since you have fairly high sample resolution, wouldn't it be possible to do a more detailed analysis similar to that done in Gfeller et al., 2014, Figure 9 (reference below)? I'm aware that you cannot assign a sample to every month, but you should be able to assign a precise age (in decimal years) to the center of each sample? When stacking over all years, you may then obtain a smooth curve (similar to Gfeller et al), that more precisely will tell you the seasonality of each impurity?

Reply: Thank you for your comment. We initially thought of using this approach, but although the temporal resolution of the core is relatively high, it is not sufficient to try a monthly or two-month resolution. For instance, the interval 1934-1933 CE is covered only by 4 samples. We used decimal years in order to plot the nssSO4 concentration records, but assigning an exact month to each ice layer would require a much higher resolution than the one we were able to provide using our decontamination procedure.

**Detailed comments**

Please be consistent in the naming of the ice core. It is sometimes called GV7 (in the title) and sometimes GV7(B) (in the abstract). Use the same notation throughout the MS unless there are several different ice cores in play?

Reply: Thank you for your comment. The name GV7 refers both to the core and to the drilling site. To avoid confusion, the text was edited so that GV7(B) was used only when referring to the core itself, while the name GV7 for the site.

In several places is mentioned 'the first meters of the core' when I think you mean 'the upper meters of the core'?

Reply: Thank you for your comment, it was edited in the text of the MS

l. 110-124: We need to know if there was a casing in the bore hole. It says that a 4 m liquid stand was 'inserted inside' (should probably be 'added to'?) the bore hole, which improved the quality of the ice core. However, we also learn that the core was full of breaks and cracks filled with contaminating drill liquid, so would it have been better to use a higher liquid stand for the drilling?

Reply: Thank you for raising this critical point. No casing was used for the bore hole and this information will be added to the manuscript. Moreover, the addition of a 4m liquid stand appeared to be a fair compromise between working time and quality of the core at that time but we agree that a different choice could have been more rewarding.

l. 125 onwards: It is stated that the isotopic analysis was done in 60 and 4 cm resolution, but I'm struggling to find out if that was also the resolution of the chemistry samples? This is important information that needs to be stated clearly.

Reply: Thank you for your comment. This information will be added to the text.

l. 278-298: Unless you are actually trying out those alternative dating methods and comparing the results, there seems to be no need to spend that much space to explain about them? 'We tried out alternative methods for layer counting (ref, ref, ref), but found that they were not suitable for our dataset for this or that reason'?

Reply: Thank you for your comment. The section will be edited as suggested, removing the methods that were tested but ultimately not used for the dating.

l. 312 onwards: You are discussing missing sections of the core. Could this be quantified, so we know how important it is and how long intervals/periods are missing?

Reply: Thank you for pointing this out. As discussed above, a section in Table 1 was added in order to display for each section of the core delimited by volcanic eruptions the amount (%) of missing core.

l. 328 onwards: Is layer thinning other than firnification taken into account here? Is flow-induced layer thinning important or can it be neglected?

Reply: Thank you for your comment. We took into consideration just the firnification thinning. The flow-induced layer thinning can be considered negligible for these reasons:

1. GV7 is located on the ice divide extending from the Oates Coast to Talos Dome

2. The ice velocity is very low (max 0.3 m yr-1, Frezzotti et al., 2007 JGR)

3. The ice thickness upstream GV7 is nearly constant, thus we expect nearly constant thinning function

Figure 2: Could you also mark where the Agung eruption is located in the profile?

Reply: Thank you for your comment, this figure will be edited and both eruptions will be marked.

Figure 3: Would be good to put an age scale or at last a few age markers in this figure similar to Figure 2. Could a similar figure be made for Agung or for any other eruptions?

Reply: Thank you for your comment; as pointed out before, this figure was removed and replaced with another one more detailed.

Figure 7: I find the break-point analysis very confusing as it does not demonstrate any coherence between the cores. Would a 50 yr smoothed curve applied to the three records not be easier to read?

Reply: the reviewer is perfectly right; our results show that there is no coherence between cores. We show that each core is representative of a different East Antarctic snow accumulation area: GV7 and Law Dome are coastal sites (~2000 km far away) whereas Talos Dome is representative of a middle plateau area (~250 km and ~2000 km far from GV7 and Law Dome, respectively). However, we will rewrite the paragraph in order to improve its readability.

**Reference:**

Gfeller, G., Fischer, H., Bigler, M., Schupbach, S., Leuenberger, D., and Mini, O.: Representativeness and seasonality of major ion records derived from NEEM firn cores, Cryosphere, 8, 1855-1870, 10.5194/tc-8-1855-2014, 2014.

---

## Author Comment (AC3)

**We thank Prof. Cole-Dai for his helpful and constructive remarks on the manuscript. Here below is reported our point-to-point reply to each stressed issue starting with "Reply".**

**"Dating of an East Antarctic ice core (GV7) by high resolution chemical stratigraphies" Nardin et al. (2021)**

**Climate of the Past**

**Public comment: Jihong Cole-Dai**

- This discussion paper details how annual layer counting is used to date an shallow ice core from a coastal region in East Antarctica. The paper gets into many specifics of how annual/seasonal cycles in measured ice core chemical content originate and are identified. This is good because these specifics are important to the accuracy and precision of ice core dating.

- Seasonal cycles in ice core parameters depend on the seasonal variations of the chemical species at their source. The Introduction section of the paper seems to imply that as along as there is a source seasonal variation, seasonal cycles can be found in ice cores. However, that is not the only dependency that may determine whether seasonal cycles can be identified and utilized for dating. Tow other factors are also important in determining whether a seasonal cycle in a paramter can be found in ice cores. One is the transport process from the source area to the ice core site. Seasonal variations in source may be moderated or even erased by differences in transport efficiency during different time periods of a year. One example is sea-salt components such as Na and Mg ions. Sea-salt aerosols may not be particularly more abundant in winter to many Antarctic locations; but stronger winds in winer and spring may bring higher levels of sea-salt aerosols to a site, resulting in a winter-high and summer-low seasonal pattern. The second factor is preservation of seasonal signals in snow. Water oxyegn isotope composition is an example: vapor diffussion can graually even out the difference between summer and winter, eventually smoothing out seasonal cycles completely. Another such example is nitrate: Even through nitrate is primarily produced in summer with photochemical reactions in the atmosphere, significant post-depositional change (reemission of nitrate as NOx) make the nitrate annual cycle in snow or ice unrecognizeable, such that nitrate is not a good annual layer indicator in in many places in Antarctica.
So, when discussing using seasonal cycles for dating in general, it may be important to point out the limitations such as those mentioned above and caution against unrealistic optimism.

Reply: Thank you for your comment and for pointing this out. The points raised by Prof. Cole-Dai are actually very relevant when studying seasonal patterns and would deserve an extensive discussion that, although very interesting, is beyond the scope of this paper. Nevertheless, we will try to summarize and briefly discuss in the revised Introduction the source, transport and preservation issues of sea salt and nitrate.

- I have some difficulty understanding the significance of the equations in (2) (lines 172-175). The first two are simply stating the facts (total is equal to the sum of sea-salt fraction and the non-sea-salt fraction). In the other two, ssNa and nssCa are supposed to be calculated from measured ion concentrations. However, calculation of ssNa requires the knowledge of nssCa, while the calculation of nssCa requires the knowledge of ssNa. How could ion measurement lead to ssNa and/or nssCa, if each requires one to know the other first?

Reply: Thank you for your comment. As previously done in other Antarctic ice cores (e.g. Roethlisberger et al., GRL 2002; Wolff et al., Nature, 2006; Benassai et al., Ann. Glaciol., 2005), to evaluate terresterial and marine contributions to $Ca^{2+}$ and $Na^+$, we just solved the equation system (2) reported in the manuscript. From IC analysis we achieve the concentration of total $Na^+$ and total $Ca^{2+}$ and, by applying the sea water and crustal ratios of these ions we derive the sea salt $Na^+$ and non-sea salt $Ca^{2+}$ content.

- A significant part of the paper or the outcome of dating the ice core by annual layer counting is an accumulation history of some 960 years. Analysis of this history by the authors shows a slight increasing trend for most parts of this period. I did not see any mention in the paper about the effect of layer thinning due to ice flow. While layer thinning in the top 10% of the ice sheet thickness is usually negligibly small, thinning in and of itself could generate or contribute to a trend, albeit small. Could the authors do a calculation based on a simple flow model to determine if layer thinning may be responsible for part of the increasing trend in accumulation? Another factor to consider is the effect of snow/ice density, for calculation of layer thickness/accumulation rate in water equivalent depends on density which could vary widely. The paper gives only a superficial description of how density was measured and how the measured density was used to be part of the calculation of annual layer thickness.

Reply: Thank you for pointing this out. Ice thinning was taken into account when calculating and discussing the snow accumulation rate. More details will be added in the main text (in the methodology section, together with further information on the density of the ice core and how it was evaluated on site), but to summarize, we assumed a constant thinning function (assuming an ice thickness of 1530 m).